# COMPOSE AND FUSE: REVISITING THE FOUNDATIONAL BOTTLENECKS IN MULTIMODAL REASONING

**Yucheng Wang**[*]   **Yifan Hou**[*]   **Aydin Javadov**   **Mubashara Akhtar**   **Mrinmaya Sachan**

{yuchwang, yifan.hou, mubashara.akhtar, mrinmaya.sachan}@inf.ethz.ch

aydin.javadov@mtec.ethz.ch

**ETH** zürich

## ABSTRACT

Multimodal large language models (MLLMs) promise enhanced reasoning by integrating diverse inputs such as text, vision, and audio. Yet, despite their perceptual strengths, their ability to reason across modalities remains underexplored, with conflicting reports on whether additional modalities help or harm performance. These inconsistencies stem from a lack of controlled evaluation frameworks and analysis of models' internals to isolate *when* and *why* modality interactions support or undermine reasoning. We address this gap through a logic-based evaluation framework that categorizes multimodal reasoning into six interaction patterns, varying how factual information is distributed across modalities and logically combined. Empirically, additional modalities enhance reasoning only when they provide independent and sufficient reasoning paths, while redundant or chained entailment support in extra modalities often hurts performance. In addition, models recognize cross-modal facts reliably and always reason on text effectively. Moreover, reasoning is degraded in three systematic ways: weaker modalities drag down overall performance, conflicts bias preference toward certain modalities, and joint signals from different modalities fail to be integrated effectively. Therefore, we identify two core failures: *task-composition bottleneck*, where recognition and reasoning cannot be jointly executed in one pass, and *fusion bottleneck*, where early integration introduces bias. For further investigation, we find that attention patterns fail to encode fact usefulness, but a simple two-step prompting (recognize then reason) restores performance, confirming the task-composition bottleneck. Moreover, modality identity remains recoverable in early layers, and softening attention in early fusion improves reasoning, highlighting biased fusion as another failure mode. In general, our findings show that integration, not perception, is the main barrier to multimodal reasoning, suggesting composition-aware training and early fusion control as promising directions.[1]

## 1 INTRODUCTION

Multimodal large language models (MLLMs) extend traditional language models beyond text to incorporate additional modalities such as vision and audio (Li et al., 2025b; Xu et al., 2025; Yu et al., 2025; Abouelenin et al., 2025). By integrating complementary signals, MLLMs can form richer and more grounded representations of the world. Text offers structured and abstract information, audio encodes temporal and prosodic signals, and images convey spatial and visual context, together capturing facets of meaning that no single modality can express in isolation (Clark & Brennan, 1991; Mayer, 2002). Through such cross-modal integration, MLLMs aim to overcome the limitations of unimodal systems, enabling more robust understanding, stronger perceptual grounding, and support for more complex reasoning (Li et al., 2023; Bie et al., 2025; Raza et al., 2025; Coburn et al., 2025).

---

[*]   Equal contribution
[1]Our code and data are publicly available.

Although MLLMs offer promise for improved reasoning by integrating diverse signals, the precise influence of additional modalities remains unclear, especially under complex reasoning scenarios. Existing studies offer conflicting observations: some report that incorporating vision or audio can improve model performance (Li et al., 2023; Guan et al., 2024; Fu et al., 2025), while others suggest that additional modalities introduce interference or confusion (Bie et al., 2025; He et al., 2025; Hou et al., 2025). However, these findings are often anecdotal or domain-specific, lacking a unified framework to systematically assess when and how multimodal input contributes to or undermines reasoning (Gupta et al., 2024; Coburn et al., 2025; Hao et al., 2025; Li et al., 2025a; Bi et al., 2025). In particular, it remains unclear under what conditions additional signals strengthen reasoning, add little, or actively impede it (Wu et al., 2025; Zhang et al., 2025). As a result, the role of modality interaction in reasoning, whether beneficial or detrimental, remains underexplored.

Moreover, most evaluations treat MLLMs as black-box systems, emphasizing external performance while leaving their internal mechanisms poorly understood (Liu et al., 2024b; Liang et al., 2023; Li et al., 2024). Even when empirical patterns emerge, such as degraded reasoning with added modalities, they are rarely accompanied by interpretability analyses that examine how models internally encode modality identity, assess evidence relevance, or perform cross-modal integration (Peng et al., 2025; Sinha et al., 2024; Yu et al., 2024; Wadekar et al., 2024). One contributing factor could lie in how these models are trained: current MLLMs are typically optimized using alignment-style objectives that pair vision or audio with text through paired supervision, contrastive learning, or instruction tuning (Zhao et al., 2024; Lin et al., 2024; Xie & Wu, 2024; Jiang et al., 2025). These objectives prioritize perceptual matching over cognitive composition, reinforcing shallow correlations rather than fostering deeper reasoning. As a result, while MLLMs often perform well on perception-heavy tasks (Li et al., 2023; Liu et al., 2024a), they struggle to generalize when reasoning demands flexible integration of multimodal information. Without interpreting the internal representations and fusion behaviors, it remains difficult to pinpoint where these limitations arise or how they might be overcome.

Previous evaluations report mixed effects of adding modalities to reasoning because they rarely control *where* decision-relevant facts appear or *how* those facts must be logically combined. We focus on *logical reasoning using information from multiple modalities*: how models use cross-modal facts to infer answers, rather than simple perception of unimodal content. To make effects measurable, we introduce six types of canonical interactions (§ 2), grounded in propositional logic, that jointly vary (i) where the crucial facts are placed across modalities and (ii) how those facts must be combined to solve the task. Instantiated through controlled, synthetic multiple choice reasoning tasks, this framework allows us to assess not only *when* added modalities that help or hurt, but also *why*.

To understand *when* added modalities help or hurt reasoning, we analyze performance across our six interactions. *Alternative* yields slight gains: added modalities help when they provide independent, individually sufficient reasoning paths. *Equivalence* offers no benefit in the presence of a strong modality (e.g., text), suggesting that redundant perceptual support rarely improves high unimodal performance. *Entailment* consistently degrades accuracy, showing that splitting multi-hop reasoning chains across modalities makes inference brittle (§ 3.2). *Independence* reveals performance bias, where reasoning accuracy is heavily dependent on which modality carries the decisive fact. *Contradictory* exposes preference bias, as the models could favor certain modalities when inputs conflict. *Complementary* highlights a weakness in fusion, where models struggle to integrate the necessary signals jointly (§ 3.3). Moreover, across all settings, text-only baselines approach a ceiling, and models reliably recognize facts across modalities. Therefore, we identify two bottlenecks: (**i**) *task composition*: models struggle to jointly perform recognition and reasoning when information is split across modalities; and (**ii**) *multi-source fusion*: models lack robust mechanisms to select, weigh and combine heterogeneous information, leading to performance, preference, and fusion biases (§ 3.4).

To explain *why* these bottlenecks arise, we probe the internal behaviors of MLLMs in a controlled setting. First, although models recognize facts well and reason effectively (in text), their internal attention patterns fail to encode *usefulness* (i.e., distinguish relevant facts from distractors). As a result, accuracy drops when recognition and reasoning must be composed in a single step, revealing a *task-composition bottleneck*. Explicitly decoupling the two stages through a two-step prompt substantially alleviates this issue (§ 4.1). Second, models preserve modality identity throughout processing, with the strongest signal concentrated in early decoder layers, but this preservation leads to biased weighting across modalities. Targeted interventions in these layers (e.g., softening early attention via increased temperature) significantly improve cross-modal reasoning, whereas modifications at later layers have little effect. This pattern confirms a *fusion bottleneck*, where biased

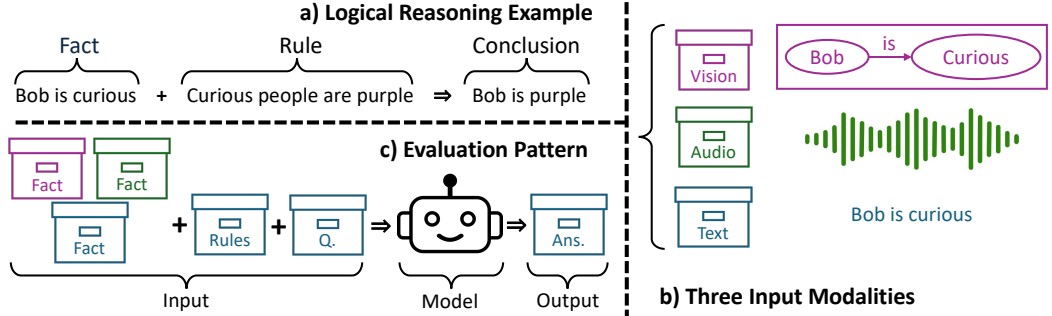

Figure 1: **Multimodal logical reasoning setup and evaluation pipeline. (a) Logical reasoning example:** a single-step deduction where the fact "*Bob is curious*" and the rule "*Curious people are purple*" entail the conclusion "*Bob is purple*." **(b) Modality renderings:** the same fact is rendered as text ("Bob is curious"), as audio via neural TTS, and as a schematic visual using graph visualization. **(c) Evaluation prompt pattern:** the model receives modality-specific fact blocks (text, audio, vision), followed by the rule set and the question with multiple-choice options; the model outputs the predicted answer.

early integration prevents the balanced use of multimodal evidence (§ 4.2). Together, these insights indicate that additional modalities yield limited and often inconsistent benefits, with failures rooted not in perception but in integration. This calls for models that incorporate composition-aware training, supervision for evidence selection, and architectural mechanisms for early fusion control, so that extra modalities become assets for reasoning rather than sources of interference.

## 2 CATEGORIZING MODALITY INTERACTIONS IN MULTIMODAL REASONING

This section introduces our evaluation framework for multimodal logical reasoning. We first describe the general setup of the task: how facts, rules, and questions are constructed and how facts are rendered across modalities (§ 2.1). We then introduce details of the six canonical interaction types, which specify how useful information is distributed across modalities (§ 2.2). These settings allow us to systematically assess whether and how models integrate, ignore, or prioritize cross-modal evidence.

### 2.1 STUDYING MULTIMODAL REASONING VIA THE LENS OF LOGICAL REASONING

Fig. 1 illustrates our reasoning task setup. Each instance consists of a set of facts, a set of rules (always in text), and a multiple-choice question. Facts convey information through different modalities: text, audio, or vision, and we control which facts are assigned to which modality (Fig. 1.b). To solve the task, the model should first correctly *recognize* the facts represented in various modality, then *reason* correctly combining with the provided rules.

**(a) Logical reasoning example.** To isolate modality interaction, we adopt a simplified single-step reasoning setting inspired by Clark et al. (2020). Full details of the data construction are in App. A.1. For example, given the fact "*Bob is curious*" and the rule "*Curious people are purple*," the model should infer "*Bob is purple*." This setup avoids multi-hop complexity and directly tests the model's ability to identify and utilize relevant information across modalities.

**(b) Three input modalities.** To minimize confounders from low-level perception, we encode each fact in three controlled modalities (see Fig. 2b): (i) a short text sentence (e.g., "Bob is curious"), (ii) audio synthesized via neural TTS,[2] and (iii) a schematic visual rendered using graph visualization.[3] These controlled renderings ensure interpretability and reduce variability due to acoustic or visual complexity, allowing us to focus on reasoning and modality integration.

**(c) Evaluation prompt pattern.** Each prompt presents a randomized set of fact blocks from different modalities, followed by the text-based rule set and a multiple-choice question. To assess robustness and bias, we also inject *noisy facts* (irrelevant distractors) into the input. In the next subsection, we

---

[2]We use CosyVoice2 TTS to convert text to speech.
[3]We use GraphViz to generate simple entity-attribute diagrams.

define six modality interaction types that vary how decision-relevant facts are distributed, enabling us to test different forms of cross-modal fusion, redundancy, and conflict.

## 2.2 INTERACTION TYPES

**Equivalence ($\equiv$).** All modalities redundantly encode the same fact (see Fig. 4 in App. A.3 for more details). For example, "*Erin is friendly*" appears in vision, audio, and text. This tests whether redundancy helps or harms reasoning, revealing models' ability to aggregate repeated evidence.

**Alternative ($\vee$).** Each modality presents a different fact, but all facts independently satisfy a disjunctive rule (Fig. 5). For example, the rule "*Friendly person is clean. Purple person is clean. Red person is clean.*" is matched by "*Erin is friendly*", "*Erin is person*", and "*Erin is red*", in separate modalities. This setting assesses whether models can leverage distinct but semantically aligned reasoning paths.

**Entailment ($\rightarrow$).** Facts are distributed across modalities to form a multi-hop reasoning chain (Fig. 6). For example, "*Erin is bouncy*" $\rightarrow$ "*Erin is bright*" $\rightarrow$ "*Erin is friendly*" $\rightarrow$ "*Erin is purple*", with each step in a different modality. Only the final fact directly supports the answer. This setting probes models' capacity for chained inference across modalities.

**Independence ($\emptyset$).** Only one modality contains the decision-relevant fact, while the others include distracting unrelated facts (Fig. 7). For example, only the vision modality presents "*Erin is friendly*," while text and audio contain irrelevant attributes. This setting tests per-modality reasoning and robustness to irrelevant signals.

**Contradictory ($\oplus$).** Each modality leads to a different conclusion (Fig. 8). For example, text implies "*Erin is clean*", vision implies "*Bob is purple*", and audio implies "*Erin is tasty*". This interaction reveals the model's default preference when faced with conflicting evidence across modalities.

**Complementary ($\wedge$).** Each modality contributes a fact required for a conjunctive rule (Fig. 9). For instance, "*Erin is friendly*", "*Erin is purple*", and "*Erin is red*", are distributed across modalities, and the rule "*If a person is friendly and purple and red, then the person is clean.*" must be applied. This interaction evaluates the ability to integrate information across modalities for multi-source reasoning.

## 3 EVALUATION OF MODALITY INTERACTIONS: PROS AND CONS

We now present a systematic evaluation of MLLMs across the six canonical interaction types. This section first outlines our experimental setup, then investigates when multiple modalities *help* or *hurt* reasoning, and finally synthesizes key bottlenecks revealed by the results.

### 3.1 PREPARATION

We begin by describing the models, prompting strategy, decoding procedure, and evaluation metric that together form a controlled testbed for analyzing modality interactions.

**Models.** We evaluated four recent open-source MLLMs that support at least three input modalities and generate text outputs. *Baichuan-Omni-1.5d (7B)* (Li et al., 2025b, Baichuan) is designed for efficient and balanced multimodal reasoning. *Qwen2.5-Omni (7B)* (Xu et al., 2025, Qwen) is a state-of-the-art model that handles text, vision, audio, and video, with streaming output, serving as a strong open baseline for multi-source reasoning. *MiniCPM-o-2.6 (8B)* (Yao et al., 2024, MiniCPM) processes text, vision, and audio, and is optimized for real-time multimodal streaming and on-device deployment. *Phi-4 Multimodal (5.6B)* (Abouelenin et al., 2025, Phi4) extends the Phi family to vision and audio, emphasizing compactness and efficiency over scale.[4] which covers both high-capacity and lightweight systems, enabling comparison of modality interaction across diverse architectures.

**Prompt design and decoding.** We use a unified prompt format across all models to ensure fair comparison. Each prompt includes a system instruction, a set of fact blocks in random modality order (text, vision, audio), a series of rules of textual reasoning, and a four-way multiple-choice question

---

[4]We refer to these models by shortened names in all experiments,

(Fig. 1c). To encourage step-by-step reasoning, we insert concise CoT hints while minimizing behavioral interference. Decoding follows HuggingFace defaults with greedy sampling to produce stable outputs, from which the final answer is automatically extracted. All models are evaluated on the same synthetic dataset using identical prompt templates. More details are provided in App. A.3.

**Evaluation metric.** We report accuracy as the primary evaluation metric. We evaluate the reasoning performance by the multiple-choice question answer format with four options, and the model's selected answer is automatically extracted from its output response. Since there are four options, random guessing would yield a baseline accuracy of 25%. To ensure robustness, each experiment is conducted on 1,000 synthetic instances per condition.

## 3.2 DO MULTIPLE MODALITIES HELP REASONING?

One motivation for using MLLMs is to enhance reasoning by incorporating additional information from multiple modalities. But does adding useful input from another modality always help? To explore this, we evaluate three controlled interaction types designed to probe distinct modes of cross-modal benefit: redundancy, optionality, and composition. The results are summarized in Tab. 1, with full breakdowns in Apps. B.1 to B.3.

**Setup.** In *Equivalence*, the same decisive fact is redundantly placed in all modalities, testing whether repetition reinforces reasoning. In *Alternative*, each modality contains a distinct but individually sufficient fact for solving the problem, allowing multiple independent reasoning paths. In *Entailment*, a reasoning chain (A → B → C → Answer) is split across modalities, requiring integration of cross-modal premises for successful inference. To assess the added value of multimodal input, we compare each multimodal setting with its unimodal baselines, where only one decisive fact is in one modality (text, vision, or audio) and the others are omitted. From this comparison, we can directly analyze the additional value brought by additional information in extra modalities.

Table 1: **Does Multimodality Help Reasoning?** Accuracy (%) and performance deltas ($\Delta$) relative to unimodal baselines across three interaction types: *Equivalence* (redundant facts across modalities), *Alternative* (independent reasoning paths), and *Entailment* (multi-hop chains split across modalities, with final-step facts in V/A/T respectively). *Alternative* settings slightly boost performance, *Equivalence* yields marginal decrease (compared to text), while *Entailment* causes notable accuracy drops on reasoning.

| Accuracy (%) | Multimodal ($\equiv, \vee, \rightarrow$) | | | | |
|---|---|---|---|---|---|
| | **Equivalence$_{\Delta V, \Delta A, \Delta T}$** | **Alternative$_{\Delta V, \Delta A, \Delta T}$** | **Entailment: V$_{\Delta V}$, A$_{\Delta A}$, T$_{\Delta T}$** | | |
| Baichuan | $84.8_{5.4\uparrow, 9.8\uparrow, 11.1\downarrow}$ | $97.6_{19.6\uparrow, 17.8\uparrow, 0.3\uparrow}$ | $79.5_{2.0\downarrow}$ | $75.6_{6.4\downarrow}$ | $80.7_{13.6\downarrow}$ |
| Qwen | $98.9_{2.6\uparrow, 4.5\uparrow, 0.9\uparrow}$ | $100.0_{3.7\uparrow, 6.1\uparrow, 2.6\uparrow}$ | $78.4_{15.7\downarrow}$ | $86.6_{8.2\downarrow}$ | $83.9_{12.8\downarrow}$ |
| MiniCPM | $94.8_{5.4\uparrow, 5.2\uparrow, 0.2\downarrow}$ | $99.1_{7.1\uparrow, 8.0\uparrow, 2.9\uparrow}$ | $81.8_{11.4\downarrow}$ | $80.0_{12.0\downarrow}$ | $88.4_{6.8\downarrow}$ |
| Phi4 | $84.1_{25.3\uparrow, 23.9\uparrow, 12.5\downarrow}$ | $97.9_{20.3\uparrow, 26.3\uparrow, 1.0\uparrow}$ | $73.0_{2.2\downarrow}$ | $69.3_{0.7\downarrow}$ | $79.7_{18.0\downarrow}$ |
| **Average** | $90.7_{9.7\uparrow, 10.9\uparrow, 5.7\downarrow}$ | $98.7_{12.7\uparrow, 14.8\uparrow, 1.7\uparrow}$ | $78.2_{7.8\downarrow}$ | $77.9_{7.1\downarrow}$ | $83.2_{12.8\downarrow}$ |

**Findings.** In *Equivalence*, models show marginal gains when the decisive fact is in vision (+9.7%) or audio (+10.9%), but performance drops when the fact is already in text (-5.7%). This suggests that redundancy is only helpful when the original modality is weak. In *Alternative*, consistent improvements are observed across all modalities (+12.7% vision, +14.8% audio, +1.7% text), indicating that semantically independent reasoning paths are successfully leveraged. However, *Entailment* leads to substantial drops in accuracy across all modalities (-7.8% vision, -7.1% audio, -12.8% text), highlighting the difficulty of cross-modal multi-hop composition.

**Observation 1.** Multimodal input improves reasoning only when it contributes additional, semantically independent reasoning paths. In contrast, redundant information provides little benefit, particularly when a strong modality (text) is already sufficient, and distributing multi-step reasoning chains across modalities often reduces accuracy. These results suggest that *the core bottleneck in multimodal reasoning lies not in recognizing facts*, since individual modalities suffice in many cases.

**Real-World Validation.** To assess whether our findings extend beyond controlled synthetic settings, we additionally evaluate the *Equivalence* interaction on IsoBench (Fu et al., 2024), a real-world vision-text benchmark containing paired representations of the same symbolic content. The results exhibit the same pattern as our synthetic experiments: multimodal (T+V) inputs provide little or no gain over strong text-only baselines. Detailed experimental setup and results are provided in App. C.

### 3.3 DO MULTIPLE MODALITIES HURT REASONING?

Although certain forms of multimodal input can aid reasoning (§ 3.2), adding modalities could also introduce errors. In this section, we evaluate three controlled settings: *Independence*, *Contradictory*, and *Complementary*: to identify specific failure modes where multiple modalities degrade reasoning.

#### 3.3.1 PERFORMANCE BIAS FOR VARIOUS MODALITIES (INDEPENDENCE)

We first explore whether models exhibit consistent reasoning *performance* across modalities.

**Setup.** In the *Independence* setting, a decisive fact appears in one modality (text, vision, or audio), while the remaining modalities contain only distractors. We compare multimodal reasoning, where facts are distributed across modalities, to unimodal baselines, where all facts (both decisive and distracting) are presented within a single modality.

**Findings.** As shown in Tab. 2 (full results can be found in App. B.4), the models perform best under the text-only condition (94.45% on average), but the accuracy drops sharply to 70.29% when facts are distributed across modalities. This is well below text-only performance but above vision-only or audio-only baselines, confirming that weaker modalities introduce noise when combined with stronger ones.

Table 2: **Performance on the *Independence* interaction.** Each instance includes one decisive fact placed in a single modality, while the others contain distractors. Multimodal reasoning accuracy falls between the best (text) and worst (vision) unimodal conditions, suggesting that modality inconsistency introduces error when aggregating information across modalities.

| Accuracy (%) | Unimodal | | | Multimodal $(\emptyset)_{\Delta V, \Delta A, \Delta T}$ |
|---|---|---|---|---|
| | **V** | **A** | **T** | |
| Baichuan | 60.2 | 72.0 | 94.8 | $67.6_{7.4\uparrow, 4.4\downarrow, 27.2\downarrow}$ |
| Qwen | 73.3 | 94.3 | 95.5 | $75.2_{1.9\uparrow, 19.1\downarrow, 20.3\downarrow}$ |
| MiniCPM | 77.6 | 83.7 | 91.2 | $78.7_{1.1\uparrow, 5.0\downarrow, 12.5\downarrow}$ |
| Phi4 | 49.9 | 48.9 | 96.3 | $59.7_{9.8\uparrow, 10.8\uparrow, 36.6\downarrow}$ |
| **Average** | 65.3 | 74.7 | 94.5 | $70.3_{5.0\uparrow, 4.4\downarrow, 24.2\downarrow}$ |

**Observation 2.** Unequal reasoning capabilities across modalities, which we refer to as *performance bias*, contribute significantly to degraded multimodal reasoning. When weaker modalities are added, they can dilute or confuse the signal from stronger ones, like text.

#### 3.3.2 PREFERENCE BIAS FOR VARIOUS MODALITIES (CONTRADICTORY)

Next, we ask whether models exhibit internal *preferences* for certain modalities.

**Setup.** In the *Contradictory* setting, each modality provides a distinct and individually sufficient reasoning path, but the answer options are mutually exclusive.[5] This setup exposes which modality a model relies on when conflicting information is presented. Crucially, preference here refers to *selection behavior under conflict*, not standalone performance.

**Results.** In Tab. 3, models show clear preference patterns: Baichuan favors vision-based answers (49.0%), Qwen tends to prefer audio (44.6%), and

Table 3: Performance on the **Contradictory** interaction, where each modality leads to a different answer. Models show clear modality preferences, highlighting inconsistent reliance on input sources.

| Answer Ratio (%) | Multimodal $(\oplus)$ | | |
|---|---|---|---|
| | **V** | **A** | **T** |
| Baichuan | 49.0 | 14.9 | 33.7 |
| Qwen | 17.2 | 44.6 | 37.6 |
| MiniCPM | 22.6 | 27.2 | 49.0 |
| Phi4 | 31.9 | 19.1 | 46.1 |

---

[5]Note that one of the four answer options is always incorrect by design, so the sum of selection ratios across modalities does not equal 100%.

both MiniCPM and Phi4 prefer text (49.0% and 46.1%). These choices are often misaligned with the models' unimodal strengths, suggesting implicit biases in modality selection under conflicting input.

**Observation 3.** In addition to performance bias, MLLMs also suffer from *preference bias*: when modalities conflict, models favor certain modalities instead of strong modalities, often inconsistently with the actual performance. This misalignment introduces further risk in multimodal reasoning.

### 3.3.3 FUSION BIAS FOR VARIOUS MODALITIES (COMPLEMENTARY)

Finally, we examine whether models can *fuse* complementary evidence across modalities when each input is necessary for inference.

**Setup.** In *Complementary* setting, each modality contains one of three facts that are jointly required to solve the reasoning task. Unlike prior settings with a single decisive fact, here all facts must be composed across modalities. We compare this condition to unimodal baselines where all three facts are provided within a one modality.

**Results.** In Tab. 4, all models perform *worse* in the multimodal setup than in any unimodal condition, even when compared to weaker modalities like vision. If biased performance is the only issue, multimodal accuracy should lie between the best and worst unimodal conditions. Instead, distributing complementary facts across modalities introduces a new failure mode: models are unable to compose multiple weak signals into a coherent reasoning chain.

Table 4: **Performance on the *Complementary* interaction.** Each modality provides one necessary fact, requiring to integrate all three to get the answer. The multimodal reasoning accuracy is lower than any unimodal condition, indicating that performance degradation stems not only from modality inconsistency but also from a true cross-modal composition bottleneck.

| Accuracy (%) | Unimodal | | | Multimodal $(\wedge)_{\Delta V, \Delta A, \Delta T}$ |
|---|---|---|---|---|
| | **V** | **A** | **T** | |
| Baichuan | 50.5 | 59.4 | 87.7 | $30.2_{20.3\downarrow,\ 29.2\downarrow,\ 57.5\downarrow}$ |
| Qwen | 87.5 | 98.8 | 98.8 | $49.9_{37.6\downarrow,\ 48.9\downarrow,\ 48.9\downarrow}$ |
| MiniCPM | 74.8 | 89.3 | 92.4 | $48.8_{26.0\downarrow,\ 40.5\downarrow,\ 43.6\downarrow}$ |
| Phi4 | 80.0 | 82.2 | 99.6 | $79.1_{0.9\downarrow,\ 3.1\downarrow,\ 20.5\downarrow}$ |
| **Average** | 73.2 | 82.4 | 94.6 | $52.0_{21.2\downarrow,\ 30.4\downarrow,\ 42.6\downarrow}$ |

**Observation 4.** MLLMs struggle to integrate complementary information across modalities, even when all inputs are individually comprehensible. This reveals that beyond biased performance and preference, there is a third failure mode in multimodal reasoning: *fusion bias*.

### 3.4 KEY BOTTLENECKS IN MULTIMODAL REASONING BASED ON OUR OBSERVATIONS

The preceding evaluations reveal consistent patterns in the way models handle multimodal reasoning. We now synthesize these findings to identify key bottlenecks in multimodal reasoning.

**Observation 5.** Across all settings, the best performance consistently comes from the text-only baseline, often approaching near-perfect accuracy. This shows that models can already perform logical reasoning reliably when inputs are centralized in a single strong modality. Combined with **Observation 1** (models can recognize facts across modalities), this indicates that the core weakness lies not in perception or reasoning in isolation, but in how these components are combined.

***Bottleneck 1*: (Recognition-Reasoning) Task Composition.** The conjunction of **Observation 1** (strong recognition) and **Observation 5** (strong unimodal reasoning) points to a first core bottleneck: *task composition*. Models falter when recognition and reasoning must be performed jointly across modalities. That is, while they can detect facts and apply reasoning rules when each task is isolated, performance drops sharply when these steps must be integrated within a single inference pass.

***Bottleneck 2*: Multi-Source (Modality) Fusion.** Observations 2–4 collectively point to a second orthogonal failure point: the inability to *fuse* information from multiple modalities in a reliable and unbiased manner. First, performance bias (**Observation 2**) shows that weak modalities dilute

reasoning when mixed with stronger ones. Second, preference bias (**Observation 3**) reveals that models often favor certain modalities in conflict, even when those modalities perform poorly in isolation. Third, fusion bias (**Observation 4**) shows that models fail to integrate complementary information spread across modalities, even when all inputs are individually comprehensible. These findings suggest that MLLMs lack robust internal mechanisms for selecting, weighing, and composing evidence from heterogeneous sources, leading to systematic failures in multimodal reasoning.

## 4    TACKLING THE IDENTIFIED MULTIMODAL REASONING BOTTLENECKS

We now investigate the underlying causes of multimodal reasoning failures by probing the internal mechanisms of MLLMs. Guided by the two key bottlenecks identified in § 3: *task composition* and *multi-source fusion*, we analyze how models represent modality and information usefulness, and whether targeted interventions can mitigate these bottlenecks.[6]

### 4.1    TASK COMPOSITION

We begin with ***Bottleneck 1***, which concerns the model's inability to compose recognition and reasoning across modalities within a single inference step. Although MLLMs can perceive facts from different modalities and reason over them in isolation, their performance degrades when these abilities must be integrated. We investigate this bottleneck by analyzing internal attention patterns and exploring whether prompting strategies can mitigate the failure.

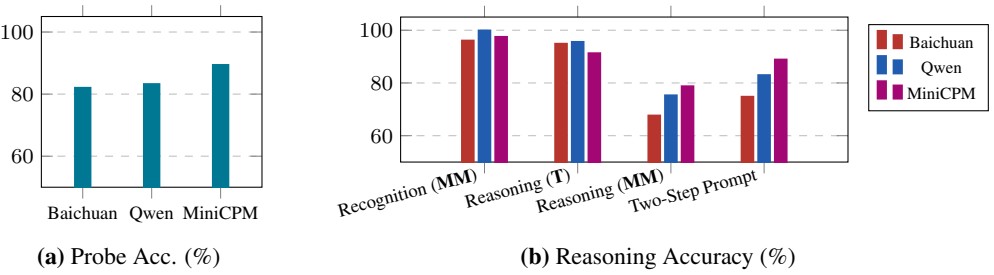

**(a)** Probe Acc. (%)          **(b)** Reasoning Accuracy (%)

Figure 2: **Attention probing and reasoning performance.** (a) Modality probing for information usefulness shows moderate accuracy, suggesting models cannot clearly distinguish useful from distractor facts. (b) Although models excel in fact recognition and text-only reasoning, their performance drops significantly on multimodal reasoning, indicating that the key limitation lies in composing recognition and reasoning across modalities.

**Probing Information Usefulness.**    We first assess whether models can internally distinguish useful facts from distractors. A linear probe is trained on decoder attention distributions: for each modality, we compute mean attention over all generated tokens and use these vectors to classify whether each fact is relevant for reasoning (more probing details are in App. A.2). The results (Fig. 2a) show that attention patterns do not reliably signal semantic usefulness, indicating that the models struggle to prioritize content relevant to decision making based on attention alone.

**Interpretation Results.**    To isolate where failures occur, we perform two controlled diagnostics: (1) a formal *recognition test*, where models identify facts across modalities without requiring additional reasoning (see Fig. 10), and (2) a *two-step prompting* setup, where recognition and reasoning are separated across prompts (see Fig. 11). As shown in Fig. 2b, the models perform nearly perfectly in recognition and maintain high accuracy in unimodal reasoning. However, when both recognition and reasoning are combined in a single multimodal prompt, accuracy drops substantially, confirming that the core failure lies in the integration of these two capabilities.

**Improving Composition via Two-Step Prompting.**    In the two-step prompting strategy, the models first extract all facts and then reason over them, substantially improving performance across all

---

[6]In this section, we conduct our analysis in the *Independence* setting to avoid multi-hop confounds and ensure interpretability. We select three representative models, Baichuan, Qwen, and MiniCPM, based on their differing modality preferences identified in § 3.3.2, which favor vision, audio, and text, respectively.

models. This shows that the failure arises not from deficiencies in perception or reasoning alone but from their joint composition within a single inference step. By decoupling these processes and providing a more explicit recognition goal, the task-composition bottleneck is effectively alleviated.

**Takeaway.** These findings highlight a core weakness in current MLLMs: despite strong perception and reasoning abilities in isolation, they lack mechanisms to integrate these steps across modalities. This reflects a broader limitation in the training objectives, which emphasizes shallow alignment rather than compositional inference. Prompt-level task decomposition offers a simple yet effective remedy.

## 4.2 MODALITY FUSION BOTTLENECK

We now address ***Bottleneck 2***, which concerns the model's difficulty in fusing information across modalities. Even when individual facts are recognized correctly, reasoning often fails due to biased or ineffective modality integration. To better understand this bottleneck, we analyze how modality identity is internally represented and whether early fusion contributes to systematic errors.

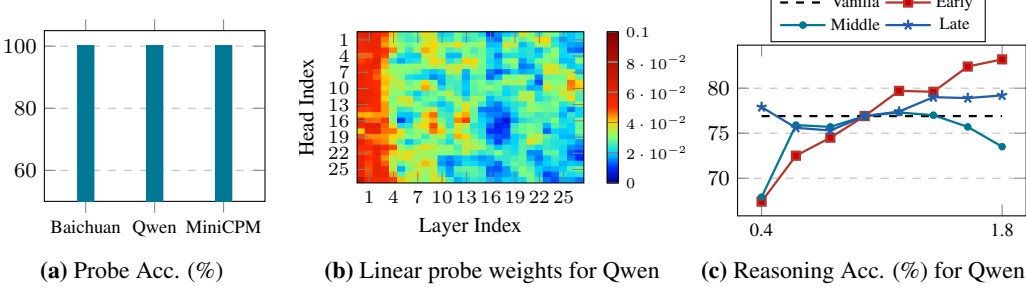

(a) Probe Acc. (%)    (b) Linear probe weights for Qwen    (c) Reasoning Acc. (%) for Qwen

Figure 3: **Modality probing based on attention patterns.** (a) All models achieve perfect probe accuracy in predicting the modality using attention patterns. (b) For Qwen, linear probe weights show that modality information is primarily captured in the first four layers. (c) Attention manipulation in different 4 layers (by adjusting head temperature from 0.4 to 1.8), where performance significantly improves in the early 4 layers.

**Probing Modality Identity.** We assess whether modality type (text, vision, audio) is preserved in the model's internal representations. A logistic regression classifier is trained on attention-derived features: for each input fact, similarly, we compute average attention from all generated tokens and flatten this into a feature vector. The classifier predicts the modality of each fact. As shown in Fig. 3a, modality identity is perfectly recoverable, indicating that even after fusion, the model maintains a strong internal signal of the input modality.

**Interpretation Results.** To locate where modality fusion occurs, we visualize layer-wise probe weights. Fig. 3b shows that the first four decoder layers carry the strongest modality signal, suggesting that fusion predominantly occurs early in the language module. Beyond this point, the modalities appear to be processed more uniformly.

**Improving Fusion via Attention Manipulation.** Motivated by this early-fusion pattern, we modify attention behavior by adjusting the softmax temperature from 0.4 to 1.8 in the first four decoder layers (Early), with the default set to 1.0. As shown in Fig. 3c, this simple intervention of increasing the temperature for the early layers yields significant improvements in reasoning accuracy by encouraging more balanced attention across the modalities. In contrast, adjusting the temperature in the middle or late layers has little effect, supporting the causal role of early fusion in the outcomes of downstream reasoning.

**Takeaways.** These findings confirm that, while the identity of the modality is well preserved, early-stage fusion introduces systematic biases that affect reasoning. A lightweight causal intervention, reshaping early attention distributions, can significantly enhance multimodal integration, highlighting the importance of fusion dynamics over perceptual bottlenecks.

**Summary of Interpretation Findings.**    Across the above interpretation analyses, we find that failures in multimodal reasoning stem not from deficiencies in perception or unimodal reasoning, but from weak *compositional integration* and *cross-modal fusion*. Although MLLMs retain modality-specific signals and can reliably recognize facts, they often default to shallow alignment behavior rather than selective integration. Without architectural biases or training objectives that explicitly encourage multimodal composition, these models remain brittle in complex reasoning tasks.

## 5    RELATED WORK

Recent benchmarks have significantly advanced the evaluation of MLLMs, measuring broad capabilities across various tasks. General-purpose suites such as MMBench (Liu et al., 2024b), MME (Fu et al., 2023), and SEED-Bench (Li et al., 2023) assess multimodal competence at scale, while reasoning-focused datasets such as MMMU (Yue et al., 2024) and robustness-oriented evaluations such as POPE (He et al., 2025) and HallusionBench (Guan et al., 2024) reveal persistent failure modes, including hallucination and modality conflicts. However, these evaluations typically measure overall performance without explicitly controlling how information is distributed across modalities, making it difficult to determine when additional modalities help, hurt, or remain redundant for reasoning.

A related line of work highlights the gap between recognition and reasoning in multimodal systems. Studies such as VERIFY (Bi et al., 2025), STARE (Li et al., 2025a), POLYMATH (Gupta et al., 2024), and EMMA (Hao et al., 2025) show that models can reliably recognize visual or symbolic elements, yet struggle when inference or structured reasoning is required. Other work examines modality interaction and dominance, for example, through visual entailment tasks (Thomas et al., 2022), qualitative taxonomies of redundancy and synergy (Sinha et al., 2024), or analyses of modality imbalance and fusion bias (Wu et al., 2025; Zhang et al., 2025). Although these studies identify important symptoms of multimodal reasoning failures, they do not provide a systematic framework that isolates the logical relationships between modalities under controlled conditions.

In contrast, our work introduces a logic-based evaluation framework that explicitly varies how facts are distributed across modalities and how they must be combined to solve a task. By defining six types of canonical interactions (e.g., equivalence, independence, complementary, contradictory) and separating recognition from reasoning, we enable direct measurement of when multimodal information improves, degrades, or fails to influence reasoning. This controlled design allows us to identify two concrete bottlenecks, task composition and modality fusion, and to validate them through targeted interventions, providing a diagnostic perspective on multimodal reasoning beyond aggregate performance evaluation.

## 6    CONCLUSION

This work presents a systematic study of how MLLMs integrate information across modalities for logical reasoning. We introduce a logic-driven evaluation framework with six canonical interaction types, allowing controlled analysis of when additional modalities help or hinder reasoning. Our results reveal a consistent pattern: modalities help only when they contribute independent, sufficient reasoning paths, while redundancy or cross-modal chaining often degrades performance. Text-only baselines already approach ceiling accuracy, underscoring that the key bottleneck is not perception but integration. Through probing and causal interventions, we identified two core bottlenecks: a *task-composition bottleneck*, where recognition and reasoning cannot be reliably combined in a single inference step, and a *fusion bottleneck*, where early-layer integration introduces modality bias. Simple remedies such as two-step prompting and attention temperature adjustments alleviate these issues, highlighting the importance of designing models and objectives that explicitly support evidence selection and unbiased fusion. We hope that our framework and findings inspire future work toward composition-aware training and architecture choices that transform added modalities from sources of interference into assets for reasoning.

## ETHICS STATEMENT

This research does not involve human participants, sensitive or private data, or applications with foreseeable negative societal impact. All datasets used are publicly available and widely used within

the vision–language and reasoning research communities. We adhere to standard best practices in data handling, model evaluation, and reporting, and our study is in full compliance with the ICLR Code of Ethics.

## REPRODUCIBILITY STATEMENT

We have taken care to make our work reproducible. The paper and appendix provide detailed descriptions of the models, datasets, and experimental setups. The code, data generation scripts, and evaluation protocols are included in the supplementary material and will be publicly available upon publication to facilitate replication and further research.

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

# A  SUPPLEMENTARY SETUP

## A.1  FACT CONSTRUCTION

We construct facts and rules following the format of Clark et al. (2020), using the data generation code from Liang et al. (2023).[7] Below we describe the details of fact generation and rule generation.

**Fact generation.**   Each fact consists of a *subject*, a *predicate*, and an *object*. The predicate is always the copula "is"; variability comes from the choice of subject and object. To keep the setting interpretable and avoid conflicts with commonsense priors, we prevent antonyms or synonyms from appearing together within the same instance. Details are as follows:

- **Subjects.** A subject is randomly sampled from three categories:
  - *Persons* (13 names): Alice, Bob, Carol, Dan, Erin, Frank, George, Harry, Iris, Jack, Kevin, Lance, Miller.
  - *Animals* (14 types): dog, cat, rabbit, mouse, tiger, lion, bear, squirrel, cow, panda, hedgehog, elephant, giraffe, hippo.
  - *Fruits* (15 types): apple, banana, orange, grape, strawberry, blueberry, watermelon, pineapple, mango, peach, cherry, pear, kiwi, lemon, plum.
- **Predicate.** Always "is" (e.g., "Bob *is* curious").
- **Objects.** Objects are adjective attributes describing the subject. We use a pool of 34 attributes: young, soft, scary, hot, smart, clean, beautiful, red, blue, green, purple, boring, strong, happy, round, big, noisy, fast, sticky, bouncy, spiky, furry, bright, shiny, magical, striped, spotted, tasty, juicy, toxic, friendly, curious, loud, sleepy.

**Rule generation.**   Rules are constructed following Liang et al. (2023), with minor modifications for brevity and formatting. Each rule specifies that one to three attribute conditions on a subject imply a new attribute. Concretely, we randomly sample attributes and instantiate rules of the form:

*If {subject} is attribute$_1$ `and/or` attribute$_2$ `and/or` attribute$_3$, then {subject} is b.*

For each instance, we generate more than four rules to ensure a sufficiently large pool of rule-grounded attributes. This enables the construction of challenging negative answer options whose attributes appear in at least one rule, preventing trivial elimination.

Following Liang et al. (2023), rules may apply either to a specific subject (e.g., "Alice") or to a subject category (e.g., "a person"), requiring the model to perform basic abstraction across individual and category-level reasoning.

Most rules contain two or three conditions. When multiple conditions are present, the conjunction is either `and` or `or`, depending on the interaction type being instantiated (e.g., `and` for Complementary interactions, `or` for Alternative interactions).

For surface realization, we adopt a compact form for single-condition rules to reduce token usage (e.g., "Weak cow is small" instead of "If a cow is weak, then the cow is small"). For multi-condition rules, we retain the explicit if–then structure to preserve clarity, e.g., "If a person is smart and young, then the person is curious".

**Image & audio generation.**   For the multimodal reasoning experiments, we require factual information in both visual and auditory formats. To generate images, we employ the Graphviz toolkit, which effectively converts structured data into clear, labeled diagrams suitable for model interpretation. For audio synthesis, we utilize CosyVoice 2 (Du et al. (2024)), one of the top Text-to-Speech (TTS) model renowned for its high consistency in timbre, achieving human-parity synthesis quality. Since the audio is solely intended to provide semantic information for subsequent logical reasoning within the MLLM, and we are not investigating the model's advanced audio understanding capabilities, we prioritize accuracy over variability in the synthesized speech. Therefore, we use CosyVoice 2's default settings to ensure the speech is clear and precise, without introducing unnecessary characteristics that could potentially interfere with the reasoning task.

---

[7]The code is from this GitHub repository.

## A.2 Experiment Setup

**LLM inference setup.** During evaluation, all models are run in float16 precision. For models that support audio output, we disable this feature and only generate text. The maximum number of generated tokens is set to 1024, which is typically sufficient for the model to produce a complete response, as long as the model does not exhibit output repetition. To ensure coherent and consistent output, we set 'do_sample=False' for all models, which is also the default setting for these models and considered best practice.

**Linear probing setup.** To construct the attention feature as the input of the probe, we extract and pool attentions. Specifically, for one data sample, the attention matrix shape should be $N \times L \times H \times O$. Here, $N$ is the number of tokens representing one fact (in one modality); $L$ is the number of layers in the model; $H$ is the number of attention head in the model; and $O$ is the number of generated tokens of the response for the model. Since $O$ and $N$ could vary in samples (or facts), we take the mean pooling in term of these two dimensions. The final attention feature for each sample is shaped as $L \times H$. We flatten the feature as one attention feature for one fact in one sample as the input for the probe classifier.

For the linear probe classifier, we use a 5-fold cross-validation setup with GroupKFold to evaluate the models. The models are trained with Logistic Regression using a regularization strength of C=1.0, an l2 penalty, and class_weight="balanced" to handle class imbalances. The feature vectors are normalized using the standard score method, which standardizes the features to have zero mean and unit variance. We apply these configurations to the task at hand. These configurations ensure a robust and consistent evaluation across all models tested.

## A.3 Prompt Templates

We list the prompt templates for all six types of interactions, the recognition task, and the two-step reasoning task. All models have the same input (template and data).

## Prompt: input and output (Equivalence)

**System:** You are an assistant tasked with solving multiple-choice questions that require logical reasoning over the supplied knowledge diagrams. Use only the information explicitly given—do not rely on outside or commonsense knowledge. Read the question and given information, think step-by-step and answer the question. At the end of your answer, answer precisely in the format 'Answer: X' where X is the chosen letter A / B / C / D.

**Input:**

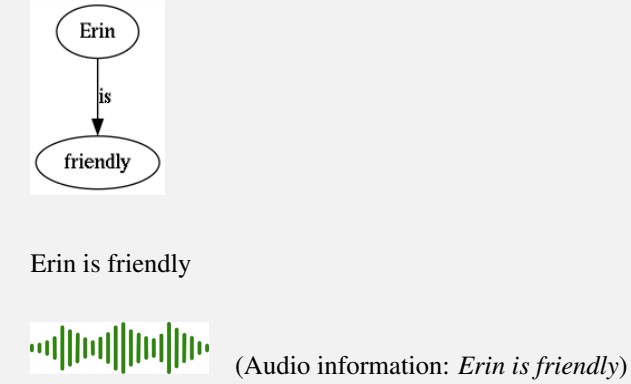

Erin is friendly

(Audio information: *Erin is friendly*)

**Rules are as follows:** Blue person is tasty. Red person is clean. If a person is smart or sleepy, then the person is curious. Friendly person is purple.

Which of the following options can be inferred based on the given facts and rules? A) Erin is curious. B) Erin is purple. C) Erin is tasty. D) Erin is clean.

**Output:** To determine which option can be inferred based on the given facts and rules, let's analyze the information step by step:

1. Given Information: - Erin is friendly.

2. Rules: - Blue person is tasty. - Red person is clean. - If a person is smart or sleepy, then the person is curious. - Friendly person is purple.

3. Analysis: - From the rule ""Friendly person is purple,"" we can infer that since Erin is friendly, Erin must be purple.

4. Checking the Options:

- A) Erin is curious. The rule does not provide a direct link between being friendly and being curious. Therefore, we cannot infer this.

- B) Erin is purple. This directly follows from the rule "Friendly person is purple."

- C) Erin is tasty. The rule does not provide a direct link between being friendly and being tasty. Therefore, we cannot infer this.

- D) Erin is clean. The rule does not provide a direct link between being friendly and being clean. Therefore, we cannot infer this.

Based on the analysis, the only option that can be inferred is:

Answer: B

Figure 4: Prompt template and model (Qwen) output (Equivalence).

## Prompt: input and output (Alternative)

**System:** You are an assistant tasked with solving multiple-choice questions that require logical reasoning over the supplied knowledge diagrams. Use only the information explicitly given—do not rely on outside or commonsense knowledge. Read the question and given information, think step-by-step and answer the question. At the end of your answer, answer precisely in the format 'Answer: X' where X is the chosen letter A / B / C / D.

**Input:**

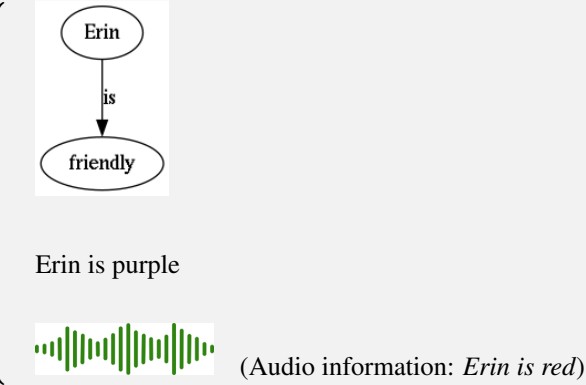

Erin is purple

(Audio information: *Erin is red*)

**Rules are as follows:** Friendly person is clean. If a person is smart or sleepy, then the person is curious. Purple person is clean. Blue Erin is tasty. Spotted Erin is beautiful. Red person is clean.

Which of the following options can be inferred based on the given facts and rules? A) Erin is clean. B) Erin is tasty. C) Erin is beautiful. D) Erin is curious.

**Output:**
To determine which option can be inferred based on the given facts and rules, let's analyze the information step by step:
1. Given Information:
- Erin is purple.
- Erin is red.
- Erin is friendly.
2. Rules: - If a person is smart or sleepy, then the person is curious. - Blue Erin is tasty. - Spotted Erin is beautiful. - Friendly person is clean. - Purple person is clean. - Red person is clean.
3. Analysis: - Since Erin is purple and red, and the rules state that purple and red people are clean, we can infer that Erin is clean. - The rules do not provide information about Erin being smart, sleepy, curious, beautiful, or tasty based on the given facts.
4. Conclusion: - The only inference we can make from the given information is that Erin is clean.
Answer: A

Figure 5: Prompt template and model (Qwen) output (Alternative).

**Prompt: input and output (Entailment)**

**System:** You are an assistant tasked with solving multiple-choice questions that require logical reasoning over the supplied knowledge diagrams. Use only the information explicitly given—do not rely on outside or commonsense knowledge. Read the question and given information, think step-by-step and answer the question. At the end of your answer, answer precisely in the format 'Answer: X' where X is the chosen letter A / B / C / D.

**Input:**

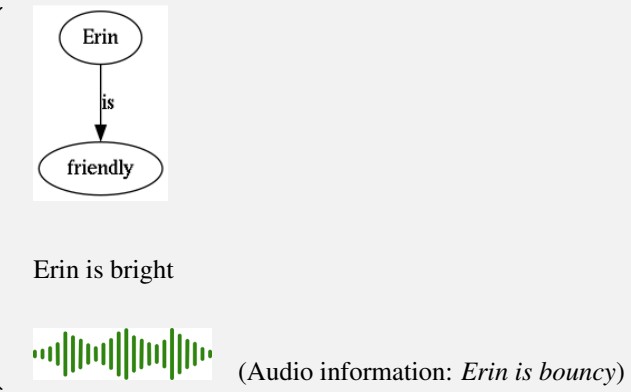

Erin is bright

(Audio information: *Erin is bouncy*)

**Rules are as follows:** If a person is smart or sleepy, then the person is curious. Red person is clean. Blue person is tasty. Bright person is friendly. Friendly person is purple. Bouncy person is bright.

Which of the following options can be inferred based on the given facts and rules? A) Erin is curious. B) Erin is tasty. C) Erin is purple. D) Erin is clean.

**Output:** Let's analyze the given information step by step:
1. Erin is bouncy.
2. Bouncy Erin is bright.
3. Erin is friendly.
4. Friendly person is purple.
5. Bright Erin is friendly.
From the information, we can deduce the following:
- Since Erin is bouncy, Erin is bright.
- Since Erin is bright, Erin is friendly.
- Since Erin is friendly, Erin is purple.
Now, let's evaluate each option:
A) Erin is curious. - This cannot be inferred because the rules do not connect being smart or sleepy to being curious.
B) Erin is tasty. - This cannot be inferred because there is no information linking being bouncy, bright, or friendly to being tasty.
C) Erin is purple. - This can be inferred because Erin is friendly, and friendly people are purple.
D) Erin is clean. - This cannot be inferred because there is no information linking being bouncy, bright, or friendly to being clean.
Answer: C

Figure 6: Prompt template and model (Qwen) output (Entailment).

## Prompt: input and output (Independence)

**System:** You are an assistant tasked with solving multiple-choice questions that require logical reasoning over the supplied knowledge diagrams. Use only the information explicitly given—do not rely on outside or commonsense knowledge. Read the question and given information, think step-by-step and answer the question. At the end of your answer, answer precisely in the format 'Answer: X' where X is the chosen letter A / B / C / D.

**Input:**

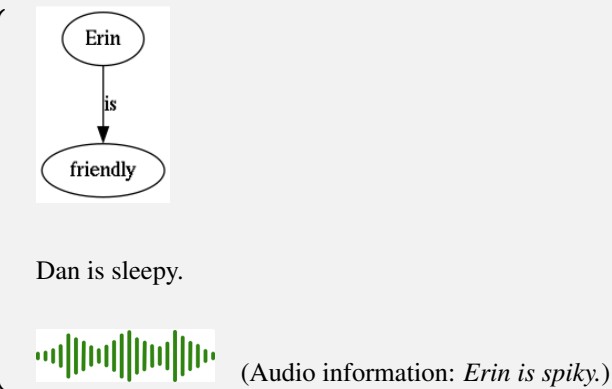

Dan is sleepy.

(Audio information: *Erin is spiky.*)

**Rules are as follows:** Blue person is tasty. Red person is clean. If a person is smart or sleepy, then the person is curious. Friendly person is purple.
Which of the following options can be inferred based on the given facts and rules? A) Erin is curious. B) Erin is purple. C) Erin is tasty. D) Erin is clean.

**Output:** To determine the correct answer, let's analyze the given information step by step:
1. Erin is friendly. 2. Friendly person is purple. 3. Blue person is tasty.
From the rules provided: - If a person is smart or sleepy, then the person is curious. - Red person is clean.
Since Erin is friendly and friendly people are purple, we can infer that Erin is purple.
Now, let's evaluate the options:
A) Erin is curious. - This cannot be inferred from the given information.
B) Erin is purple. - This can be inferred from the given information.
C) Erin is tasty. - This cannot be inferred from the given information.
D) Erin is clean. - This cannot be inferred from the given information.
Answer: B

Figure 7: Prompt template and model (Qwen) output (Independence).

## Prompt: input and output (Contradictory)

**System:** You are an assistant tasked with solving multiple-choice questions that require logical reasoning over the supplied knowledge diagrams. Use only the information explicitly given—do not rely on outside or commonsense knowledge. Read the question and given information, think step-by-step and answer the question. At the end of your answer, answer precisely in the format 'Answer: X' where X is the chosen letter A / B / C / D.

**Input:**

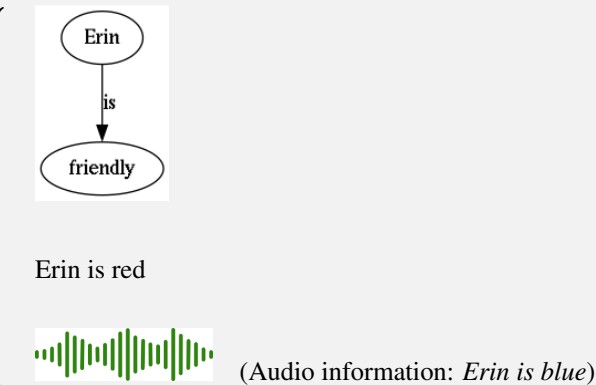

Erin is red

(Audio information: *Erin is blue*)

**Rules are as follows:** Blue person is tasty. Red person is clean. If a person is smart or sleepy, then the person is curious. Friendly person is purple.

Which of the following options can be inferred based on the given facts and rules? A) Erin is curious. B) Erin is tasty. C) Erin is purple. D) Erin is clean.

**Output:**
Let's analyze the information step by step:
1. Erin is blue.
2. Blue people are tasty.
3. Erin is friendly.
4. Friendly people are purple.
From the rules, we can infer the following:
- Since Erin is blue, she is tasty.
- Since Erin is friendly, she is purple.
Now, let's evaluate each option:
A) Erin is curious. - This cannot be inferred because the rules do not connect being blue or friendly to being curious.
B) Erin is tasty. - This can be inferred because blue people are tasty.
C) Erin is purple. - This can be inferred because friendly people are purple.
D) Erin is clean. - This cannot be inferred because the rules do not connect being blue or friendly to being clean.
Based on the given facts and rules, the options that can be inferred are B and C. However, since the question asks for a single answer, we need to choose the one that is directly supported by the rules without additional assumptions.
Answer: B

Figure 8: Prompt template and model (Qwen) output (Contradictory).

## Prompt: input and output (Complementary)

**System:** You are an assistant tasked with solving multiple-choice questions that require logical reasoning over the supplied knowledge diagrams. Use only the information explicitly given—do not rely on outside or commonsense knowledge. Read the question and given information, think step-by-step and answer the question. At the end of your answer, answer precisely in the format 'Answer: X' where X is the chosen letter A / B / C / D.

**Input:**

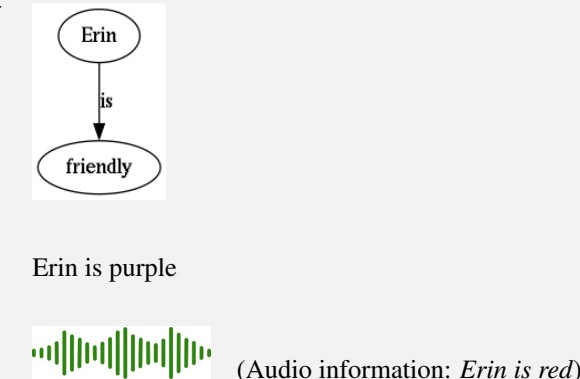

Erin is purple

(Audio information: *Erin is red*)

If a person is purple and red and not friendly, then the person is soft. If a person is friendly and purple and not red, then the person is big. If a person is friendly and red and not purple, then the person is scary. If a person is friendly and purple and red, then the person is clean.

Which of the following options can be inferred based on the given facts and rules? A) Erin is soft. B) Erin is scary. C) Erin is clean. D) Erin is big.

**Output:**

To determine which option can be inferred based on the given facts and rules, let's analyze the information step by step:
1. Erin is purple.
2. Erin is friendly.
3. Erin is red.
Now, let's apply the rules to Erin:
- Rule 1: If a person is purple and red and not friendly, then the person is soft. - Erin is purple and red and friendly, so this rule does not apply to Erin.
- Rule 2: If a person is friendly and purple and not red, then the person is big. - Erin is friendly, purple, and red, so this rule does not apply to Erin.
- Rule 3: If a person is friendly and red and not purple, then the person is scary. - Erin is friendly, red, and purple, so this rule does not apply to Erin.
- Rule 4: If a person is friendly and purple and red, then the person is clean. - Erin is friendly, purple, and red, so this rule applies to Erin.
Based on the analysis, the only rule that applies to Erin is Rule 4, which states that if a person is friendly and purple and red, then the person is clean.
Answer: C

Figure 9: Prompt template and model (Qwen) output (Complementary).

**Prompt: input and output (Recognition)**

**System:** You are an assistant tasked with solving multiple-choice questions about knowledge diagrams. Use only the information explicitly given—do not rely on outside or commonsense knowledge. The facts are given in image, audio and text. Read the question and given information, and directly answer the question in the following format: 'Answer: X' where X is the chosen letter A / B / C / D.

**Input:**

Erin

is

friendly

Dan is sleepy.

(Audio information: *Erin is spiky.*)

Question: Which fact is mentioned in the given information in image, audio or text?
Options: A) Erin is sticky., B) Erin is friendly., C) Erin is scary., D) Erin is green.

**Output:**
B

Figure 10: Prompt template and model (Qwen) output for the fact recognition.

## Prompt: input and output (Two-Step Reasoning: Step 1)

**System Prompt 1:** You are an assistant tasked with solving multiple-choice questions about knowledge diagrams. Use only the information explicitly given—do not rely on outside or commonsense knowledge. The facts are given in image, audio and text. Read the question and given information, and directly answer the question.

**System Prompt 2:** You are an assistant tasked with solving multiple-choice questions that require logical reasoning over the supplied knowledge diagrams. Use only the information explicitly given—do not rely on outside or commonsense knowledge. Read the question and given information, think step-by-step and answer the question. At the end of your answer, answer precisely in the format 'Answer: X' where X is the chosen letter A / B / C / D.

**Input 1:**

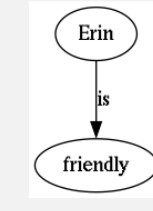

Dan is sleepy.

(Audio information: *Erin is spiky.*)

Question: Check the given information and list all the facts in the given image, audio and text, respectively.

**Output 1:**
Facts from the image: - Erin is spiky.
Facts from the audio: - Dan is sleepy.
Facts from the text: - Erin is friendly.

**Prompt: input and output (Two-Step Reasoning: Step 2)**

**Input 2 (Including Input 1 + Output 1):**
Rules are as follows: Blue person is tasty. Red person is clean. If a person is smart or sleepy, then the person is curious. Friendly person is purple.
Which of the following options can be inferred based on the given facts and rules? A) Erin is curious. B) Erin is purple. C) Erin is tasty. D) Erin is clean.

**Output 2:**
Let's analyze the information step by step:
1. Erin is friendly.
2. Erin is spiky.
3. Dan is sleepy.
From the rules provided:
- If a person is smart or sleepy, then the person is curious. - Friendly person is purple.
- Blue person is tasty. - Red person is clean.
Since Erin is friendly, according to the rules, Erin must be purple.
Now, let's evaluate the options:
A) Erin is curious. - We don't have enough information to infer this from the given facts and rules.
B) Erin is purple. - This is directly inferred from the rule that a friendly person is purple.
C) Erin is tasty. - We don't have enough information to infer this from the given facts and rules.
D) Erin is clean. - We don't have enough information to infer this from the given facts and rules.
Answer: B

Figure 11: Prompt template and model (Qwen) output for the two step reasoning.

# B SUPPLEMENTARY RESULTS

## B.1 MODALITY INTERACTION: EQUIVALENCE

**Setup.** We now evaluate the *Equivalence* interaction, where the same decisive fact is redundantly placed in all three modalities (vision, audio, text). The unimodal conditions serve as baselines, where the only fact is presented in one modality. This setting examines whether redundant presentation across modalities can reinforce fact recognition and improve reasoning robustness.

Table 5: **Performance on the *Equivalence* interaction.** Each modality redundantly contains the same decisive fact. Results show that adding equivalent facts from other modalities provides no consistent improvement, suggesting that unimodal fact recognition is not the reasoning bottleneck.

| Accuracy (%) | Unimodal | | | Multimodal $(\equiv)_{\Delta V, \Delta A, \Delta T}$ |
|---|---|---|---|---|
| | **V** | **A** | **T** | |
| Baichuan | 79.4 | 75.0 | 95.9 | $84.8_{5.4\uparrow, 9.8\uparrow, 11.1\downarrow}$ |
| Qwen | 96.3 | 94.4 | 98.0 | $98.9_{2.6\uparrow, 4.5\uparrow, 0.9\uparrow}$ |
| MiniCPM | 89.4 | 89.6 | 95.0 | $94.8_{5.4\uparrow, 5.2\uparrow, 0.2\downarrow}$ |
| Phi4 | 58.8 | 60.2 | 96.6 | $84.1_{25.3\uparrow, 23.9\uparrow, 12.5\downarrow}$ |
| **Average** | 91.0 | 79.8 | 96.4 | $90.7_{9.7\uparrow, 10.9\uparrow, 5.7\downarrow}$ |

**Results.** As shown in Tab. 5, adding equivalent facts in vision and audio brings little to no gain over the text-only setting, which already achieves near-perfect accuracy. In some cases (e.g., Baichuan and Phi4), the multimodal setting even leads to a significant performance drop. This suggests that additional modalities do not enhance reasoning and may introduce unnecessary interference. Since models already recognize text-based facts effectively, improving recognition robustness does not translate into better multimodal reasoning. The bottleneck appears to lie elsewhere, specifically, in how the recognized facts are composed during reasoning.

**Takeaway.** Adding redundant evidence across modalities fails to improve reasoning and can even hurt performance. This suggests that text-based fact recognition is not the limiting factor. Instead, cross-modal fusion and composition seem to be the main challenges in multimodal reasoning. *Since simply reinforcing facts does not help, we next ask: can additional modalities help by introducing new and independent reasoning paths?* We investigate this in the following setting.

## B.2 MODALITY INTERACTION: ALTERNATIVE

**Setup.** In this setting, each modality contains a unique antecedent that is individually sufficient to trigger the same rule and yield the correct conclusion. Thus, the model can arrive at the correct answer by reasoning over any one of the modalities. This evaluates whether models can flexibly leverage semantically diverse cues when multiple independent reasoning paths are available.

Table 6: **Performance on the Alternative interaction.** Each modality provides an independent reasoning path, and the correct answer can be inferred from any one of them. Multimodal accuracy improves slightly over the text-only baseline, showing that models can benefit from semantically diverse cues across modalities. This suggests that introducing complementary reasoning paths can help mitigate modality-specific limitations.

| Accuracy (%) | Unimodal | | | Multimodal $(\vee)_{\Delta V, \Delta A, \Delta T}$ |
|---|---|---|---|---|
| | **V** | **A** | **T** | |
| Baichuan | 78.0 | 79.8 | 97.3 | $97.6_{19.6\uparrow, 17.8\uparrow, 0.3\uparrow}$ |
| Qwen | 96.3 | 93.9 | 97.4 | $100.0_{3.7\uparrow, 6.1\uparrow, 2.6\uparrow}$ |
| MiniCPM | 92.0 | 91.1 | 96.2 | $99.1_{7.1\uparrow, 8.0\uparrow, 2.9\uparrow}$ |
| Phi4 | 77.6 | 71.6 | 96.9 | $97.9_{20.3\uparrow, 26.3\uparrow, 1.0\uparrow}$ |
| **Average** | 86.0 | 83.9 | 97.0 | $98.7_{12.7\uparrow, 14.8\uparrow, 1.7\uparrow}$ |

**Results.** As shown in Tab. 6, models achieve strong performance across unimodal settings, particularly in the text-only case. Notably, when all three sufficient premises are presented across modalities, performance consistently improves across models. For example, Qwen improves from 97.4% (text-only) to 100.0%, and MiniCPM improves from 96.2% to 99.1%. Although the gain is modest compared to the text baseline, the upward trend suggests that models can effectively integrate multiple alternative cues to reinforce the reasoning process, even when distributed across modalities.

**Takeaway.** These findings suggest that unlike redundant (Equivalence) setups, providing alternative reasoning paths across modalities can support more robust reasoning. Multimodal information helps when it offers diverse routes to the same conclusion, rather than simply reiterating or fragmenting the required information. Since introducing alternative single-step reasoning cues across modalities leads to measurable gains, it raises a natural follow-up question: can models also benefit from more indirect, multi-hop cues, such as entailment chains spread across modalities? We explore this next.

### B.3    MODALITY INTERACTION: ENTAILMENT

**Setup.** In the Entailment interaction, the model must perform multi-hop reasoning through a chain of rules. Specifically, three facts form a reasoning chain: $A \rightarrow B \rightarrow C$, where the final answer is entailed by $C$. While the decisive fact $C$ is always placed in one fixed modality (vision, audio, or text), the earlier support facts $A$ and $B$ are distributed across the other two modalities. This setting evaluates whether the model can integrate indirect, cross-modal evidence to support the final-step reasoning.

Table 7: **Performance on the Entailment interaction.** Each modality carries a fact needed for multi-hop reasoning, with the final-step premise fixed in one modality (V/A/T). Models consistently perform worse in multimodal setups compared to their unimodal baselines. This suggests that spreading multi-hop reasoning steps across modalities introduces substantial integration errors, regardless of where the final step is placed.

| Accuracy (%) | Unimodal | | | Multimodal ($\rightarrow$): Final-Step Fact | | |
|---|---|---|---|---|---|---|
| | **V** | **A** | **T** | $\mathbf{V_{\Delta V}}$ | $\mathbf{A_{\Delta A}}$ | $\mathbf{T_{\Delta T}}$ |
| Baichuan | 81.5 | 82.0 | 94.3 | $79.5_{2.0\downarrow}$ | $75.6_{6.4\downarrow}$ | $80.7_{13.6\downarrow}$ |
| Qwen | 94.1 | 94.8 | 96.7 | $78.4_{15.7\downarrow}$ | $86.6_{8.2\downarrow}$ | $83.9_{12.8\downarrow}$ |
| MiniCPM | 93.2 | 92.9 | 95.2 | $81.8_{11.4\downarrow}$ | $80.0_{12.9\downarrow}$ | $88.4_{6.8\downarrow}$ |
| Phi4 | 75.2 | 70.0 | 97.7 | $73.0_{2.2\downarrow}$ | $69.3_{0.7\downarrow}$ | $79.7_{18.0\downarrow}$ |
| **Average** | 86.0 | 84.9 | 96.0 | $78.2_{7.8\downarrow}$ | $77.9_{7.1\downarrow}$ | $83.2_{12.8\downarrow}$ |

**Results.** As shown in Tab. 7, all models experience significant performance drops in the multimodal condition compared to their unimodal counterparts, regardless of which modality carries the final-step fact. For instance, when the final-step fact is in text, we observe up to 12.8% drop compared to the text-only baseline. This pattern holds consistently across models and configurations. These results suggest that the cross-modal composition of reasoning chains introduces substantial integration errors, even when the decisive premise remains in a strong reasoning modality.

**Takeaway.** Unlike alternative information, which offers parallel reasoning paths, entailment information provides indirect, chained support, which proves to be not only unhelpful but actively harmful to reasoning performance. This highlights that current MLLMs struggle with cross-modal multi-hop reasoning, even when all required facts are present. We have now evaluated three types of auxiliary information: equivalence, alternative, and entailment, and found that only alternative reasoning paths offer modest improvements. In contrast, redundant or indirect information often introduces confusion and leads to performance degradation.

### B.4    MODALITY INTERACTION: INDEPENDENCE

Tab. 8 presents detailed results under the *Independence* interaction, where each instance contains a single decisive fact placed in one modality (vision, audio, or text), while the other modalities contain only irrelevant distractors. We evaluate both unimodal reasoning (all facts are given in one modality) and multimodal reasoning (the decisive fact is mixed with distractors across modalities).

Table 8: **Performance on the *Independence* interaction with cross-modal distractors.** Each instance contains one decisive fact placed in a specific modality (V: vision, A: audio, T: text), while the other modalities contain only noisy facts. We report accuracy (%) for each unimodal condition as well as multimodal reasoning when decisive facts are distributed.

| Accuracy (%) | Unimodal | | | Multimodal (Decisive Facts) | | | |
|---|---|---|---|---|---|---|---|
| | V | A | T | V | A | T | Random |
| Baichuan | 60.2 | 72.0 | 94.8 | 74.3 | 53.5 | 74.9 | 67.6 |
| Qwen | 73.3 | 94.3 | 95.5 | 50.8 | 90.8 | 84.1 | 75.2 |
| MiniCPM | 77.6 | 83.7 | 91.2 | 66.8 | 78.2 | 91.0 | 78.7 |
| Phi4 | 49.9 | 48.9 | 96.3 | 58.0 | 50.4 | 70.7 | 59.7 |
| **Average** | 65.3 | 74.7 | 94.5 | 62.5 | 68.2 | 80.2 | 70.3 |

Across all models, text consistently yields the highest unimodal accuracy (average 94.45%), while vision and audio vary significantly in performance. Interestingly, when the decisive fact remains in a fixed modality but is surrounded by irrelevant facts from other modalities, we observe substantial performance drops for vision and audio (e.g., Qwen drops from 73.3% to 50.8% when adding distractors to vision). The "Random" column averages over all cross-modal settings with randomly selected decisive modalities, showing that even a single irrelevant modality can degrade reasoning.

## C  VALIDATION ON REAL-WORLD ISOMORPHIC REPRESENTATIONS (ISOBENCH)

To verify that our findings are not artifacts of synthetic data, we conduct a partial validation on real-world isomorphic representations using IsoBench (Fu et al., 2024). IsoBench provides paired text and image inputs that encode the same underlying symbolic information (e.g., mathematical expressions, graphs, and board states), enabling evaluation of the *Equivalence* interaction in natural visual formats.

Although existing real-world benchmarks do not support all six interaction types in our framework, IsoBench allows us to replicate the *Equivalence* setting for the vision–text modality pair. Using the same models as in our main experiments, we evaluate unimodal text (T), unimodal vision (V), and multimodal text+vision (T+V) inputs under identical prompting and evaluation protocols.

Table 9: **Performance on IsoBench under the *Equivalence* interaction.** We report accuracy (%) for unimodal vision (V), unimodal text (T), and multimodal text+vision (T+V).

| Accuracy (%) | V | T | T+V$_{\Delta V, \Delta T}$ |
|---|---|---|---|
| Baichuan | 58.6 | 67.1 | 61.5$_{2.9\uparrow, 5.6\downarrow}$ |
| Qwen | 57.6 | 68.1 | 66.2$_{8.6\uparrow, 1.9\downarrow}$ |
| MiniCPM | 58.1 | 63.3 | 63.7$_{5.6\uparrow, 0.4\uparrow}$ |
| Phi4 | 60.3 | 66.2 | 63.1$_{2.8\uparrow, 3.1\downarrow}$ |
| **Average** | 58.7 | 66.2 | 63.6$_{4.9\uparrow, 2.6\downarrow}$ |

As shown in Tab. 9, we observe the same pattern as in our synthetic experiments: redundant multimodal inputs provide little or no improvement over strong text-only baselines, and in some cases slightly degrade performance. This consistency indicates that the task-composition and modality-fusion bottlenecks identified in our framework extend beyond controlled synthetic settings to real-world isomorphic representations.

## LLM USAGE

We used ChatGPT as a general-purpose assistant in preparing this paper. In particular, LLMs were employed for grammar refinement, clarity improvements, LaTeX formatting, and debugging minor

code snippets. They were not involved in research ideation, experimental design, or the development of theoretical contributions.

