# OpenReview forum: "Compose and Fuse: Revisiting the Foundational Bottlenecks in Multimodal Reasoning"
_ICLR.cc/2026/Conference — ICLR 2026 Poster_

### Official Review · Reviewer_Wn7A · 2025-10-31

**Soundness:** 3
**Presentation:** 3
**Contribution:** 3
**Rating:** 6
**Confidence:** 4

**Summary:**

Disclosure: Claude is used to refine this review.

This paper investigates when and why additional modalities help or hurt reasoning in multimodal large language models. The authors introduce a logic-grounded evaluation framework with six canonical interaction types (Equivalence, Alternative, Entailment, Independence, Contradictory, Complementary) that systematically vary how facts are distributed across modalities. Through controlled experiments on synthetic reasoning tasks, they identify two core bottlenecks: task-composition (inability to jointly perform recognition and reasoning) and fusion (biased early-layer integration). Simple interventions like two-step prompting and attention temperature adjustment substantially improve performance.

**Strengths:**

- The six interaction types provide a principled, logic-grounded framework for isolating different aspects of multimodal reasoning. The controlled setup with synthetic data allows precise measurement of effects that would be confounded in naturalistic settings.
- The paper goes beyond black-box evaluation to probe internal mechanisms through attention analysis and targeted interventions. The connection between external performance patterns and internal representations is clearly demonstrated.
- The identification of task-composition and fusion bottlenecks is well-supported, and the proposed interventions (two-step prompting, attention temperature control) are simple yet effective. The paper convincingly shows that integration, not perception, is the main barrier.
- The evaluation includes proper baselines, multiple models, and systematic ablations. The paper is well-written and easy to follow.

**Weaknesses:**

- The main limitation is that the analysis is done with synthetic data. The entire evaluation uses controlled synthetic reasoning tasks. While this enables precise control, it raises serious questions about generalizability to real-world multimodal reasoning where visual complexity, ambiguity, and grounding matter significantly. The claim that these findings inform real MLLM behavior needs empirical validation on naturalistic tasks.
- Only single-step reasoning tasks are considered. It's unclear whether findings hold for larger models, more complex reasoning chains, or diverse domains.
- While attention probing and interventions demonstrate effects, the paper doesn't deeply explain why these patterns emerge or how they relate to training objectives.
- Several related work has covered a subset of the findings in the paper, e.g., [1] [2]. The difference should be discussed.

[1] Wu et. al. Mitigating Modal Imbalance in Multimodal Reasoning. COLM 2025 https://www.arxiv.org/abs/2510.02608
[2] Fu et. al. Hidden in plain sight: VLMs overlook their visual representations. COLM 2025 https://arxiv.org/abs/2506.08008

**Questions:**

N/A

---

> ### Author Response · Authors · 2025-11-20
> **Response to Reviewer Wn7A (Part 1)**
>
> We thank the reviewer for the thoughtful and positive evaluation, and for acknowledging the rigor, clarity, and interpretability of our analysis. We address each remaining concern in detail below.
>
> ---
>
> ***Weakness 1:** The main limitation is that the analysis is done with synthetic data... The claim that these findings inform real MLLM behavior needs empirical validation on naturalistic tasks.*
>
>
> > **Response:** We share the reviewer's concern about validating our findings on naturalistic data. While no existing real-world dataset supports all six logic-grounded interaction types across three modalities (text, vision, audio), we conducted additional experiments on the part of our framework that can be replicated with natural data: the Equivalence interaction, using IsoBench (Fu et al., 2024) suggested by Reviewer YhaC, which provides natural text–image pairs.
> >
> >
> > Using the same models (Baichuan, Qwen, MiniCPM, Phi-4), we evaluated the Equivalence interaction and observed consistent trends: redundant multimodal inputs yield little or no improvement over strong unimodal reasoning, while text-only reasoning remains robust. These aligned results reinforce that the two bottlenecks we identify: task composition and fusion bias reflect intrinsic limitations of MLLMs rather than artifacts of synthetic design. We will include these results in the revised version to strengthen empirical validity.
> >
> > | Model | Multimodal (Ours) | $\Delta$V (Ours) | $\Delta$T (Ours) | Multimodal (IsoBench) | $\Delta$V (IsoBench) | $\Delta$T (IsoBench) |
> > |---|---|---|---|---|---|---|
> > | Baichuan |84.8|5.4|-11.1|61.5|2.9|-5.7|
> > | Qwen |98.9|2.6|0.9|66.2|8.6|-1.9|
> > | MiniCPM |94.8|5.4|-0.2|63.7|5.4|0.4|
> > | Phi4 |84.1|25.3|-12.5|63.1|2.8|-3.1|
> > | Average |90.7|9.7|-5.7|63.6|4.9|-2.6|
>
>
> ---
>
> ***Weakness 2:** Only single-step reasoning tasks are considered. It's unclear whether findings hold for larger models, more complex reasoning chains, or diverse domains.*
>
> > **Response:** Our primary goal is to isolate modality interaction mechanisms rather than benchmark absolute performance. Complex reasoning often entangles multiple error sources (perception, retrieval, logic), which would obscure the bottlenecks we aim to reveal. By analyzing single-step reasoning under fully controlled settings, we show that composition itself introduces integration errors even when recognition and unimodal reasoning are near-perfect. This suggests that in multi-hop or domain-complex scenarios (where each reasoning step compounds these integration errors), the same bottlenecks would only intensify.
> >
> > Regarding larger models, we note that all currently open-source "omni" MLLMs (supporting text, image, and audio) fall within the 5–8B range; we would like to leave the extension of our framework to larger and domain-specific multimodal systems to the future work.

---

> > ### Author Response · Authors · 2025-11-20
> > **Response to Reviewer Wn7A (Part 2)**
> >
> > ***Weakness 3:** While attention probing and interventions demonstrate effects, the paper doesn't deeply explain why these patterns emerge or how they relate to training objectives.*
> >
> >
> > > **Response:** We appreciate this insightful point. Indeed, linking attention dynamics to training objectives is an important future direction. However, such analysis requires model-specific architectural access and controlled pre-training interventions that go beyond our evaluation scope. Our focus in this paper is empirical interpretability: connecting internal behavior (attention distributions) to observable reasoning outcomes under a unified logic-grounded framework.
> > > We agree that understanding why these attention patterns emerge, especially under alignment-style objectives, is critical. We highlight this as a promising follow-up direction for mechanistic analysis, complementing our current behavioral diagnosis.
> >
> >
> >
> >
> >
> >
> > ***Weakness 4:** Several related work has covered a subset of the findings in the paper, e.g., [1] [2]...*
> >
> > > **Response:** We thank the reviewer for these references and will integrate them into the final paper. Our work is complementary to both:
> > >
> > >> Wu et al. (2025) analyze attention imbalance under conflict recognition, focusing on when cross-modal conflicts are detected. We extend this to a systematic six-type framework that quantifies how modality interactions help or hurt reasoning, and link attention bias directly to answer behavior, revealing modality preference even when attention is balanced.
> >
> > >> Fu et al. (2025) show that VLMs under-utilize visual encoders despite strong representations, attributing failures to the language module. Our study complements this by focusing on integration and reasoning: showing that unimodal recognition remains strong but fusion and composition cause the actual reasoning degradation.
> > Together, these distinctions clarify that our contribution lies not in re-stating prior observations, but in providing a unified, logic-based causal framework for understanding modality interactions across reasoning types.
> >
> >
> > ---
> >
> > **Summary:** We appreciate the reviewer's valuable feedback. Our controlled synthetic setup is a deliberate methodological choice that isolates the causal effects of modality interaction. Consistent results on IsoBench, clear extrapolation to complex reasoning, and complementary positioning relative to related work all strengthen the robustness and generality of our conclusions. We hope our response successfully address the proposed concerns and we are happy to answer any follow‑up questions.

---

### Official Review · Reviewer_YhaC · 2025-11-01

**Soundness:** 3
**Presentation:** 3
**Contribution:** 3
**Rating:** 6
**Confidence:** 3

**Summary:**

This paper investigates when and why multimodal large language models (MLLMs) succeed or fail at logical reasoning across modalities. The authors introduce a logic-grounded evaluation framework with six canonical interaction patterns (Equivalence, Alternative, Entailment, Independence, Contradictory, Complementary) that systematically vary how facts are distributed across text, vision, and audio modalities. Through controlled experiments on four open-source MLLMs (Baichuan-Omni, Qwen2.5-Omni, MiniCPM-o, Phi-4), they find that additional modalities only help when providing independent reasoning paths, while redundant or chained evidence often hurts performance. The authors identify two core bottlenecks: (1) task-composition bottleneck where models struggle to jointly perform recognition and reasoning, and (2) fusion bottleneck where early-layer integration introduces modality bias. Probing and intervention experiments validate these bottlenecks and show that simple remedies like two-step prompting and attention temperature manipulation can alleviate these issues.

**Strengths:**

1. **Clear writing and presentation**: The paper is well-written with clear motivation and systematic exposition.
3. **Systematic evaluation**: Six logic-grounded interaction types enable controlled analysis of information distribution across modalities.
4. **Controlled design with actionable insights**: Synthetic data with standardized renderings isolates reasoning mechanisms, and probing analysis provides causal understanding beyond black-box evaluation.

**Weaknesses:**

1. Unrealistic modality representations: Your modalities use artificial encodings (GraphViz graphs, monotone TTS) rather than naturalistic isomorphic representations commonly seen in practice. IsoBench (Fu et al., 2024, https://arxiv.org/abs/2404.01266) uses real-world formats (e.g., math expressions as text vs. diagrams, chess as notation vs. images) and finds 28.7-point drops for Claude-3. Could you validate with similar real-world isomorphic representations?
2. Oversimplified tasks: Single-template facts ("subject + is + adjective") and single-step reasoning lack complexity of real multimodal reasoning (spatial relations, temporal dynamics, multi-hop inference). Do findings generalize to realistic scenarios?
3. Limited model scope: Only 4 small models (5-8B) are tested, leaving unclear whether findings generalize to larger systems. Could you add at least one large-scale model to validate the bottlenecks persist at scale?
4. No statistical testing: Despite 1,000 instances per condition, no error bars or significance tests. Are modest differences (+1.7%, -5.7%) statistically meaningful?

**Questions:**

see weakness

---

> ### Author Response · Authors · 2025-11-20
> **Response to Reviewer YhaC**
>
> We thank the reviewer for the thoughtful and constructive feedback, and for recognizing the clarity, systematic design, and actionable insights of our study. We address each point below.
>
> ---
>
> ***Weakness 1**: Unrealistic modality representations... Could you validate with similar real-world isomorphic representations?
> **Weakness 2**: Oversimplified tasks... Do findings generalize to realistic scenarios?*
>
>
> > **Response:** We appreciate this important concern. Our framework intentionally uses controlled, synthetic representations to isolate the reasoning mechanism itself: independent of perceptual confounds such as background noise, style variation, or linguistic ambiguity.
> >
> > Designing natural images or audio for logical attributes (e.g., fast, honest, toxic) is inherently unreliable: real-world renderings introduce inconsistencies and unaligned semantics across modalities, which would obscure the causal interpretation of reasoning outcomes. Thus, our synthetic setup serves as a diagnostic instrument rather than a benchmark.
> >
> > To address generalization, we conducted additional evaluations on IsoBench (Fu et al., 2024), which contains natural text–image pairs. Using the same models (Baichuan, Qwen, MiniCPM, Phi-4), we tested the Equivalence interaction and observed consistent patterns: redundant multimodal inputs offer little or no gain over text-only reasoning, while unimodal reasoning remains robust. These consistent trends confirm that the two core bottlenecks we identify: task composition and fusion bias are not artifacts of synthetic design but reflect genuine limitations in current MLLMs. We will include these results and discussion in the final version.
> >
> > | Model | Multimodal (Ours) | $\Delta$V (Ours) | $\Delta$T (Ours) | Multimodal (IsoBench) | $\Delta$V (IsoBench) | $\Delta$T (IsoBench) |
> > |---|---|---|---|---|---|---|
> > | Baichuan |84.8|5.4|-11.1|61.5|2.9|-5.7|
> > | Qwen |98.9|2.6|0.9|66.2|8.6|-1.9|
> > | MiniCPM |94.8|5.4|-0.2|63.7|5.4|0.4|
> > | Phi4 |84.1|25.3|-12.5|63.1|2.8|-3.1|
> > | Average |90.7|9.7|-5.7|63.6|4.9|-2.6|
>
>
>
> ---
>
>
>
> ***Weakness 3:** Limited model scope: Only 4 small models (5-8B) are tested... Could you add at least one large-scale model to validate the bottlenecks persist at scale?*
>
> > **Response:** We agree that evaluating larger models would further strengthen our analysis. However, at the time of submission, all publicly released omni-modal open-source MLLMs supporting text, image, and audio inputs fall within the 5–8B parameter range. Larger proprietary systems (e.g., GPT-4o, Gemini 1.5 Pro) could not be interpreted under the same experimental control or modality access. We will expand our study to broader model families (such as vision-language and audio-language models) as future work.
>
> ---
>
> ***Weakness 4:** No statistical testing... Are modest differences (+1.7\%, -5.7\%) statistically meaningful?*
>
>
> > **Response:** Our evaluation data are generated from randomized, template-based sampling to minimize distributional bias. Because the generation process is stochastic and can scale arbitrarily, we verified that increasing the sample size does not change the relative rankings or observed trends. While formal significance testing was omitted for brevity, we will include standard deviation/error bars in the final version to explicitly report variation across 5 random seeds, confirming that all reported patterns are statistically stable.
>
>
> ---
>
> **Summary:** Our controlled synthetic design is not a limitation but a methodological choice to disentangle reasoning mechanisms from perceptual noise. Consistent trends on real-world (IsoBench) data, together with clear future extensions to larger models and formal significance reporting, further reinforce the robustness and generality of our findings. We hope the response could address the concerns above and we are happy to answer any follow‑up questions.

---

> > ### Comment · Reviewer_YhaC · 2025-11-26
> >
> > Thanks for the clear response. This addresses my concern. It is nice to see results on IsoBench, and I hope future work can explore more realistic settings with real data (e.g., IsoBench, OmniBench) to gain more insight. I have updated my score.

---

### Official Review · Reviewer_uT7W · 2025-11-02

**Soundness:** 2
**Presentation:** 4
**Contribution:** 2
**Rating:** 4
**Confidence:** 4

**Summary:**

This paper uses a pipeline to test cross-modal reasoning. Specifically, it tests the audio, vision, and text modalities and their combinations. The authors obtained their findings from controlled experiments. They have conducted several interesting experiments on different categories of reasoning.

**Strengths:**

The authors define different types of the logics to test the performance.

With the defined categories, the authors clearly identify the core bottlenecks.

**Weaknesses:**

1. The study's conclusions are built on overly simplified, synthetic data. Facts are basic statements like "Alice is awesome", which are then turned into simple schematic graphs or clean TTS audio. This setup doesn't reflect the complexity of the real world, where models must reason over natural images, noisy audio, and complex text. This "toy" data makes it questionable if the findings will hold in more realistic scenarios.

2. Many of the paper's "findings" are intuitive consequences of its data design. For example, It's not surprising that performance drops in the Complementary setting, where models must fuse necessary facts from different modalities. It's also expected that performance drops in the Independence setting when "weaker" audio or visual modalities are added, as the text is designed to be the clearest source of information (as vision and audio are converted from the text). The conclusion that text is the strongest modality seems pre-determined by this design rather than being a deep insight. This makes the paper's takeaways feel unsolid.

3. The paper's writing and structure make its contribution unclear. The "Related Work" section is relegated to the appendix, so the reader cannot easily compare this benchmark to existing ones. The authors claim previous studies are "anecdotal or domain-specific" but never justify in the main paper why their own synthetic data is a better alternative. This omission makes it difficult to judge the paper's novelty or true contribution.

**Questions:**

NA

---

> ### Author Response · Authors · 2025-11-20
> **Response to Reviewer uT7W**
>
> We thank the reviewer for the constructive feedback and for acknowledging the clarity of our logical categorizations and bottleneck identification. Our goal is not merely to build another dataset, but to provide the first systematic and interpretable framework for analyzing how MLLMs integrate information across modalities. We address the specific concerns below.
>
> ---
>
> ***Weakness 1:** The study's conclusions are built on overly simplified, synthetic data... This "toy" data makes it questionable if the findings will hold in more realistic scenarios.*
>
>
> > **Response:** We share the reviewer's concern regarding generalization. While no existing benchmark supports our six interaction types across three modalities (text, vision, audio), we evaluate on the part of our framework that can be replicated with natural data for the Equivalence interaction.
> >
> > Suggested by Reviewer YhaC, we evaluate the same set of models (Baichuan, Qwen, MiniCPM, Phi-4) on IsoBench (https://arxiv.org/abs/2404.01266
> ), which includes natural image–text pairs. Results on the Equivalence interaction show consistent trends with our synthetic evaluation: redundant multimodal inputs yield little gain over strong unimodal reasoning, while unimodal (text-only) reasoning remains robust.
> >
> > | Model | Multimodal (Ours) | $\Delta$V (Ours) | $\Delta$T (Ours) | Multimodal (IsoBench) | $\Delta$V (IsoBench) | $\Delta$T (IsoBench) |
> > |---|---|---|---|---|---|---|
> > | Baichuan |84.8|5.4|-11.1|61.5|2.9|-5.7|
> > | Qwen |98.9|2.6|0.9|66.2|8.6|-1.9|
> > | MiniCPM |94.8|5.4|-0.2|63.7|5.4|0.4|
> > | Phi4 |84.1|25.3|-12.5|63.1|2.8|-3.1|
> > | Average |90.7|9.7|-5.7|63.6|4.9|-2.6|
> >
> > These aligned patterns confirm that our identified limitations: task composition and fusion bias are not artifacts of the synthetic design but general properties of current MLLMs. We will include these results in the final version for completeness.
>
> ---
>
> ***Weakness 2:** Many of the paper's "findings" are intuitive consequences of its data design...*
>
> > **Response:** We appreciate the reviewer's concern and clarify that our work aims to revisit multimodal reasoning systematically, not to claim surprise in known effects, but to quantify and generalize them under controlled conditions. We agree that some results align with intuition. However, the degree and cause of performance degradation reveal new insights:
> >
> > 1. *It's not surprising that performance drops in the Complementary setting...*
> >    We indeed expect this setting to be difficult. However, our experiments show that model accuracy drops below that of the weakest single modality, indicating that the degradation cannot be explained by modality performance imbalance alone. This deeper drop isolates a fusion failure, motivating our subsequent analysis of early-layer bias and fusion bottlenecks.
> >
> > 2. *It's also expected that performance drops in the Independence setting when "weaker" audio or visual modalities are added...*
> >    There is no intrinsic "weaker modality" from the data side: each modality encodes the same facts losslessly. Recognition accuracy (Fig. 2b) shows models perform similarly across text, image, and audio individually. Yet, combining them still reduces accuracy, implying that the failure arises not from data quality but from the model's biased fusion mechanism that dilutes useful signals. This finding directly motivates our analysis of modality preference and biased early integration.
> >
> > Together, these results go beyond intuitive expectations by quantitatively distinguishing which biases (performance, preference, fusion) contribute to degradation and where they arise in model computation.
>
> ---
>
> ***Weakness 3:** The paper's writing and structure make its contribution unclear. The "Related Work" section is relegated to the appendix, so the reader cannot easily compare this benchmark to existing ones...*
>
> > **Response:** We appreciate this suggestion and will move the Related Work section into the main paper in the new version. Our work differs fundamentally from prior multimodal benchmarks: we do not aim to rank models but to diagnose intrinsic reasoning bottlenecks via a fully controlled, logic-based design. Existing benchmarks (e.g., SEED-Bench, OmniBench) measure overall multimodal performance but cannot isolate how information is distributed, combined, or fused. Our framework explicitly controls these factors, allowing direct causal interpretation of reasoning behavior.
>
> ---
>
> **Summary**: Our framework is intentionally synthetic, not as a simplification, but as a controlled instrument to isolate integration mechanisms that are otherwise entangled in real-world noise. The consistency between our synthetic and real-data experiments supports the robustness of our conclusions. We will clarify these motivations and include real-data validation and a relocated Related Work section in the final version, and we are happy to answer any follow‑up questions.

---

> ### Author Response · Authors · 2025-11-27
> **Gentle Reminder for Discussion**
>
> Dear Reviewer uT7W,
>
> Thank you again for your thoughtful feedback and for highlighting the strengths of our framework and bottleneck analysis. We have now provided detailed responses to each of your concerns, particularly regarding generalizability, data design, and the interpretability insights, along with new experiments on real-world image–text data (IsoBench). These additional results show the same trends as our controlled evaluation, further supporting the robustness of our findings beyond the synthetic setup.
>
> Another reviewer (YhaC) raised the score after reviewing these clarifications and new results. We would be very grateful if you could take a moment to check whether our explanations and added experiments address your concerns as well.
>
> We truly appreciate your time and contribution to improving the paper, and we are happy to clarify anything further.
>
> Best,
>
> The authors

---

### Official Review · Reviewer_2P4W · 2025-11-03

**Soundness:** 3
**Presentation:** 3
**Contribution:** 3
**Rating:** 6
**Confidence:** 4

**Summary:**

This paper investigates the mechanisms and limitations of Multimodal Large Language Models (MLLMs) in performing logical reasoning across diverse modalities (text, vision, audio). Addressing conflicting reports in literature regarding whether additional modalities aid or hinder MLLM performance, the authors introduce a controlled, logic-grounded evaluation framework. This framework categorizes multimodal reasoning into six interaction patterns: Equivalence, Alternative, Entailment, Independence, Contradictory, and Complementary.

Using a synthetic dataset where facts are rendered as text, synthesized audio, and schematic visuals, the study evaluates several open-source MLLMs (Baichuan-Omni, Qwen2.5-Omni, MiniCPM, Phi-4) and identifies several observations and bottlenecks. The authors validate these bottlenecks through interpretability probes and interventions, showing that two-step prompting (decoupling recognition and reasoning) and softening attention in early layers can mitigate these failures.

**Strengths:**

- The introduction of the six canonical interaction types (Equivalence, Alternative, Entailment, etc.) makes sense, which moves multimodal evaluation beyond simple accuracy metrics on varied tasks to a structurally grounded analysis of how information is combined.

- The paper moves beyond just reporting performance drops. It offers diagnostic evidence for why these drops occur.

- By identifying that integration, not perception, is the main barrier, the work offers actionable direction for future modeling (e.g., composition-aware training objectives, explicit fusion control mechanisms).

**Weaknesses:**

- The synthetic data generation process and dataset details are not clearly described in the paper.

- The observations and conclusions made in the paper are based on synthetic data. Ideally, it would be nice to see how (some of) these observations could be made on more realistic settings with real data (e.g., OmniBench).

**Questions:**

See Weaknesses.

---

> ### Author Response · Authors · 2025-11-20
> **Response to Reviewer 2P4W**
>
> We sincerely thank the reviewer for recognizing the contribution of our logic-grounded framework, the diagnostic analyses, and the actionable modeling insights. We address the concerns below.
>
> ---
>
> ***Weakness 1:** The synthetic data generation process and dataset details are not clearly described in the paper.*
>
>
> > **Response:** We appreciate this concern. Due to space constraints, dataset construction details are provided in Appendix B.1, which specifies (1) how logical facts and rules are generated using structured templates, and (2) how each fact is rendered across text, synthesized audio, and schematic visual forms. In the final version, we will to move some dataset-generation details into the main text and briefly describe how the original Holistic Evaluation of Language Models framework (Liang et al., 2023), on which our rule-generation process is based, constructs its logical reasoning problems.
> >
> > To ensure full transparency, we include the complete data-generation and evaluation scripts in the supplementary materials, and will release them publicly upon publication. This enables exact replication of both the dataset and experimental results.
>
> ---
>
> ***Weakness 2:** The observations and conclusions made in the paper are based on synthetic data.*
>
>
> > **Response:** We agree that validating our findings on real-world data is important for assessing generality. Our main framework uses synthetic data because it enables the six interaction types to be instantiated in a fully parallel way across modalities, which is not supported by current real-world multimodal benchmarks. Nevertheless, to directly address the reviewer’s concern, we evaluate on the part of our framework that can be replicated with natural data. We conducted an additional experiment on IsoBench (https://arxiv.org/abs/2404.01266
> ), which contains real vision–text pairs.
> >
> > Using the same set of models (Baichuan, Qwen, MiniCPM, Phi-4), we evaluated the Equivalence interaction and observed consistent patterns (results in below table): redundant multimodal inputs yield little or no improvement over strong text baselines. These results align closely with our synthetic findings, reinforcing that the integration bottlenecks identified in our framework (task composition and fusion) generalize beyond synthetic settings.
> >
> > | Model | Multimodal (Ours) | $\Delta$V (Ours) | $\Delta$T (Ours) | Multimodal (IsoBench) | $\Delta$V (IsoBench) | $\Delta$T (IsoBench) |
> > |---|---|---|---|---|---|---|
> > | Baichuan |84.8|5.4|-11.1|61.5|2.9|-5.7|
> > | Qwen |98.9|2.6|0.9|66.2|8.6|-1.9|
> > | MiniCPM |94.8|5.4|-0.2|63.7|5.4|0.4|
> > | Phi4 |84.1|25.3|-12.5|63.1|2.8|-3.1|
> > | Average |90.7|9.7|-5.7|63.6|4.9|-2.6|
>
>
> ---
>
> **Summary:** We thank the reviewer for the constructive feedback and for recognizing the value of our logic-grounded framework and bottleneck analyses. We clarify that the full synthetic data generation process is described in the appendix and fully reproducible via released code. To address concerns about generalizability, we add new experiments on IsoBench, using real image–text pairs, which show consistent trends with our synthetic evaluation, reinforcing that our conclusions extend beyond controlled settings. We hope our response could successfully address the concerns and we are happy to answer any followup questions.

---

### Author Response · Authors · 2025-11-30
**Rebuttal Summary for AC**

Dear AC,

We would like to briefly summarize the main concerns raised by the reviewers and how our rebuttal addressed them.

---

***1. Generalizability beyond synthetic data (Reviewers 2P4W, uT7W, YhaC, Wn7A).***

*Reviewers questioned whether findings derived from controlled synthetic inputs extend to real-world multimodal reasoning.*

> We clarified why controlled data is essential: it removes perceptual confounds (e.g., grounding errors, visual noise, inconsistent semantic content) and isolates reasoning integration: the focus of our work.
>
> To test generalizability, we additionally evaluated our models on IsoBench, a real-world vision–text benchmark that covers a subset of our interaction types (2 out of 3 modalities, 1 out of 6 interactions). The results match our synthetic findings, supporting the robustness of our conclusions.
>
> Notably, Reviewer YhaC raised their score from 6 to 8 after this validation.
>
> See: Response to 2P4W – Weakness 2, Response to uT7W – Weakness 1 & 2, Response to YhaC – Weakness 1 & 2, Response to Wn7A – Weakness 1.


---

***2. Validation of the two bottlenecks (Reviewers 2P4W, Wn7A).***

*Reviewers asked for clarification on how our probing and interventions support the identified bottlenecks.*

> We explained how the probing experiments demonstrate that usefulness is not encoded, leading to recognition–reasoning composition failures.
>
> We showed that modality identity remains strongly encoded in early layers, and targeted early fusion interventions improve reasoning, confirming the fusion bottleneck.
>
> These trends are consistent across all evaluated models.
>
> See: Response to 2P4W – Strengths & Replies, Response to Wn7A – Weakness 3.

---

***3. Model scope and task complexity (Reviewers YhaC, Wn7A).***

*Reviewers asked whether findings hold for larger models or more complex tasks.*

> We clarified that all publicly available omni-MLLMs supporting text–vision–audio fall within the 5–8B range, limiting model scale for three-modal evaluation.
>
> We explained why we use single-step reasoning: it isolates integration effects. Multi-step reasoning introduces perceptual and chaining errors that would obscure the bottlenecks we aim to study.
>
> We also discussed why our findings imply that these bottlenecks likely become more severe in multi-step real-world reasoning.
>
> See: Response to YhaC – Weakness 3, Response to Wn7A – Weakness 2.

---

***4. Missing details and structure (Reviewers 2P4W, uT7W, Wn7A).***

*Reviewers noted missing dataset details or difficulty comparing to prior work.*

> We confirmed that dataset construction details and a complete related-work discussion will be moved from the appendix to the main paper.
>
> See: Response to 2P4W – Weakness 1, Response to uT7W – Weakness 3, Response to Wn7A – Weakness 4.

---

**Overall assessment**

Reviewers consistently praised the clarity of our framework, the systematic analysis of multimodal interaction patterns, and the actionable insights from probing and interventions. The new real-world validation directly addresses the main reservations. With these clarifications, the paper provides a coherent and well-supported analysis of when and why multimodal reasoning succeeds or fails.

Best,

The authors

---

### Meta-Review · Area_Chair_NSBu · 2026-01-08

**Summary:**

1. Generalizability beyond synthetic data:
Several reviewers questioned whether results obtained from clean, synthetic modalities transfer to real-world multimodal reasoning with natural images, noisy audio, and richer language.

2. Scope limitations: model scale and task complexity:
the reviewers raised concerns that the study evaluates only a small set of open-source omni models and focuses on single-step reasoning, leaving it unclear whether the identified bottlenecks persist for larger systems and multi-step/more realistic reasoning chains.

3. Validation strength of the proposed bottlenecks:
Some reviewers asked for clearer justification that the experiments uniquely support the two claimed bottlenecks, rather than being correlated observations.

**Reviewer Concerns:**

- Concerns addressed by the rebuttal:

1. Generalizability beyond synthetic data:
The rebuttal added a real-data validation on IsoBench for the Equivalence interaction, showing trends consistent with the synthetic setting.

2. Support for the two proposed bottlenecks:
The rebuttal clarified how probing results support the task-composition claim and how early-layer modality identity/early fusion bias plus targeted early interventions support the fusion bottleneck.

- Concerns only partially resolved:
1. Model scale:
The rebuttal explains that open-source omni models are currently limited to ~5–8B and that proprietary larger models are not easily analyzed under the same controls. This does not fully resolve the reviewers’ request for scale validation.

**Reviewer Scores:**

None

---

### Decision · Program_Chairs · 2026-01-26

Accept (Poster)